# An Update on Phytochemicals and Pharmacological Activities of the Genus *Persicaria* and *Polygonum*

**DOI:** 10.3390/molecules26195956

**Published:** 2021-10-01

**Authors:** Gisela Seimandi, Norma Álvarez, María Inés Stegmayer, Laura Fernández, Verónica Ruiz, María Alejandra Favaro, Marcos Derita

**Affiliations:** 1ICiAgro Litoral, CONICET, Facultad de Ciencias Agrarias, Universidad Nacional del Litoral, Kreder 2805, Esperanza 3080HOF, Argentina; giselaseimandi@hotmail.com.ar (G.S.); nalvarez@fca.unl.edu.ar (N.Á.); mistegmayer@gmail.com (M.I.S.); laurafernandez1@gmail.com (L.F.); mfavaro@fca.unl.edu.ar (M.A.F.); 2Farmacognosia, Facultad de Ciencias Bioquímicas y Farmacéuticas, Universidad Nacional de Rosario, Suipacha 531, Rosario S2002LRK, Argentina

**Keywords:** natural products, *Persicaria*, *Polygonum*, infectious diseases, bioactive compounds, pharmacological activities

## Abstract

The discovery of new pharmaceutical identities, particularly anti-infective agents, represents an urgent need due to the increase in immunocompromised patients and the ineffectiveness/toxicity of the drugs currently used. The scientific community has recognized in the last decades the importance of the plant kingdom as a huge source of novel molecules which could act against different type of infections or illness. However, the great diversity of plant species makes it difficult to select them with probabilities of success, adding to the fact that existing information is difficult to find, it is atomized or disordered. *Persicaria* and *Polygonum* constitute two of the main representatives of the Polygonaceae family, which have been extensively used in traditional medicine worldwide. Important and structurally diverse bioactive compounds have been isolated from these genera of wild plants; among them, sesquiterpenes and flavonoids should be remarked. In this article, we firstly mention all the species reported with pharmacological use and their geographical distribution. Moreover, a number of tables which summarize an update detailing the type of natural product (extract or isolated compound), applied doses, displayed bioassays and the results obtained for the main bioactivities of these genera cited in the literature during the past 40 years. Antimicrobial, antioxidant, analgesic and anti-inflammatory, antinociceptive, anticancer, antiviral, antiparasitic, anti-diabetic, antipyretic, hepatoprotective, diuretic, gastroprotective and neuropharmacological activities were explored and reviewed in this work, concluding that both genera could be the source for upcoming molecules to treat different human diseases.

## 1. Introduction

Infectious diseases are caused by pathogenic microorganisms such as bacteria, viruses, parasites or fungi and can be transmitted, directly or indirectly, from one person to another. In the last decades, the rapid population growth, poverty, urban migration, international travels and environment changes have increased the exposure to several infectious agents [1]. According to the World Health Organization’s estimations, infectious diseases constitute one-third of all deaths in the world. Despite the pharmaceutical efforts to design new antibiotics for the treatment of these diseases, the acquisition of microorganisms’ resistance represents a worldwide concern, and this is attributed to the indiscriminate and improper use of current antimicrobial drugs [2]. For this reason, there is a need to search for alternative anti-infective substances, for example, antimicrobials of plant origin.

Plants have developed different active principles for defense towards the pathogens’ attacks, called secondary metabolites. Phenolic compounds (e.g., coumarines and phytoalexins), terpenoids (e.g., monoterpenes, saponins, steroids) and nitrogen compounds (e.g., alkaloids and lectins) are some secondary metabolites with biocidal capacity against human pathogens [3,4]. It has been estimated that 14–28% of higher plant species are used medicinally [5]. The use of antimicrobials of plant origin has many advantages [6]: they have no secondary effects, better patient tolerance, are less expensive, increase the bioavailability of free agents and demonstrate adequate therapeutic effect with relatively small doses.

The genus *Polygonum* comprises 250 species (20 represented in Argentina) of annual and perennial herbs distributed from the northern temperate to tropical and subtropical regions, preferably in low and humid areas, on the banks of streams and rivers. Nowadays, these species are classified into the *Persicaria* and *Polygonum* genera, according to botanical and phytochemical characteristics. Particularly, flavonoids and sesquiterpenes have played an important role in the systematics of *Polygonum* species as a chemotaxonomic marker and contributed to the regrouping of these species into *Persicaria* and *Polygonum* genera [7]. Species of *Persicaria* and *Polygonum* show different biological properties for, e.g., antiseptic, antibiotic [8], antinociceptive [9], antifungal [8,10,11], diuretic, antirheumatic, astringent [12,13] uses, and for the treatment of external infections such dermatoses, ulcers and sores [13,14].

The selection of the manuscripts for this review was based on the following inclusion criteria: articles published in English in the last three decades, with the keywords *Persicaria*, *Polygonum* and anti-infective in the title, abstract or full text. The authors firstly selected articles according to the title, then to the abstract and then through an analysis of the publication full text. The resulting articles were manually reviewed with the goal of identifying and excluding the works that did not fit the criteria described above. For scientific names of species, The Plant List updated database nomenclature was followed (www.theplantlist.org (accessed on 1 October 2021)).

## 2. Pharmacological Uses and Phytochemical Composition of *Persicaria* and *Polygonum* Species

The *Persicaria* and *Polygonum* genera are known for their wide spectra of properties to treat different diseases (Table 1). This is possible due to the capacity to produce a great variety of secondary metabolites such sesquiterpenes containing dialdehydes as functional groups [15]; flavonoids [7,16]; neoflavonoids [17]; triterpenes [18,19]; lignans [20]; other types of phenolic compounds not included among flavonoids subclass [21]; phenylpropanoids [22,23]; phenolic compounds belonging to the tannins type [24]; coumarins [17,18]; and anthraquinones [23].

## 3. Pharmacological Activities of *Persicaria* and *Polygonum* Products: Different Type of Extracts and Compounds Responsible for the Bioactivities

The following paragraphs will summarize, mainly in table format, the major biological activities depicted in the literature for these plant genera, not only for the different type of extracts generated from them but also for the pure compounds isolated. The parts of each species used to obtain the bioactive phytochemicals and a summary of the results, thrown by the displayed bioassays, will be also comment.

### 3.1. Antimicrobial Activity

Many authors all over the world have investigated the ability of *Persicaria* and *Polygonum* species to treat human fungal and bacterial pathogens. Table 2 summarizes the plants which show antimicrobial capacities against different fungal and bacterial strains.

From the analysis of Table 1, it could be stated that the three main bioassays displayed for detecting antimicrobial activities of different type of extracts of *Polygonum* or *Persicaria* species, as well as their bioactive compounds, include the percentage of microorganism growth inhibition, the determination of IC_50_ or the Minimum Inhibitory Concentration of each tested sample (extract or isolated compound). The last one constitutes a standard method for detecting antimicrobial susceptibility, so that the results obtained from carrying out these bioassays should be more convincing. In this sense, *P. acuminata*, *P. ferruginea*, *P. hydropiperoides*, *P. lapathifolia* and *P. arenastrum* were tested only as antifungals against yeasts or filamentous fungi, resulting in *P. acuminata* and *P. ferruginea* being the most active ones (MICs between 3.9 and 125 µg/mL). Authors correlated these promising activities with the presence of the sesquiterpene polygodial in *P. acuminata* extracts [25] and the chalcones cardamonin and pashanone present in *P. ferruginea* extracts [8]. Moreover, *P. chinensis*, *P. hydropiper*, *P. maculosa*, *P. punctata*, *P. senegalensis*, *P. aviculare* and *P. cognatum* resulted in both antifungals and antibacterials, highlighting the MICs values obtained for isolated compounds confertifolin and drimenol from *P. hydropiper* essential oil and polygodial from its chloroformic extract (MICs between 0.39 and 125 µg/mL comparable with standard drugs) [71,72,73,74,75]. Finally, *P. capitata*, *P. glabra*, *P. minor*, *P. tinctoria* and *P. perfoliatum* were reported in the literature only as antibacterials, remarking the high activity against S. aureus, E. coli, K. pneumoniae and N. gonorrhoeae of flavonoid-enriched fractions of aqueous extracts of *P. capitata* (MICs comparables to ciprofloxacin) [48]. These results are promising but limited to the in vitro evaluation, and thus, more studies regarding solubilities, absorption, blood distribution, pharmacodynamics, pharmacokinetics and tissue toxicity should be performed before these extracts or compounds could become remedies.

### 3.2. Antioxidant Activity

Species of the *Persicaria* and *Polygonum* genera have a remarkable antioxidant activity, as some compounds can remove the excess of free radicals in bodies to maintain normal metabolisms. Table 3 summarizes the most interesting results found in the literature for the following antioxidant bioassays performed: DPPH (2,2-diphenyl-2-picryl hydroxyl); TEAC (Trolox Equivalent Antioxidant Capacity); CUPRAC (Cupric Reducing Antioxidant Capacity); ABTS ((2,2′-azino-bis(3-ethylbenzothiazoline-6-sulfonic acid)); NBT (Nitroblue tetrazolium); FRAP (Ferric Reducing Antioxidant Power); ORAC (Oxygen Radical Absorbance Capacity assay); and CCA (Copper Chelating Activity).

From the analysis of this table, it could be remarked that gallic, chlorogenic and ellagic acids may be the responsible for the strong antioxidant activities showed by *P. equisetiforme*, *P. lapathifolia*, *P. amplexicaulis*, *P. chinense* var. *chinense*, *P. chinense* var. *hispidum*, *P. bellardii*, *P. paleaceum* and *P. sagittata*, which in many experiments resulted to be even more potent than the standard drugs. On the other hand, some specific compounds such as persilben, amplexicine, quercitrin and brevifolin carboxylic acids, vanicoside A and B and taxifolin, were associated to the antioxidant response during the different bioassays carried out with *P. maculosa*, *P. amplexicaulis*, *P. chinense* var. *chinense*, *P. chinense* var. *hispidum*, *P. sagittata* and *P. orientalis*, respectively. Conspicuously, a study performed with MeOH extract and zinc oxide nanoparticles (ZnO-NPs) of *P. bistorta* inhibited the ABTS radicals with an IC_50_ value of 40 µg/mL, and it was observed that the activity was dose-dependent. Moreover, it is well known that phenolic compounds are widespread in the plant kingdom, acting as antioxidants offering plant protections, so the *Persicaria* and *Polygonum* genera are not exceptional.

### 3.3. Analgesic and Anti-Inflammatory Activity

Many species of the *Persicaria* and *Polygonum* genera present anti-inflammatory properties, which are described in Table 4.

From the analyses in Table 4, it could be observed that unlike the bioactivities described in the previous tables, for the evaluation of analgesic and anti-inflammatory effects, the whole extract of each species was evaluated more than the isolated compounds. Oppositely, more in vivo studies were informed for these bioactivities. The MeOH extract of *P. chinensis* exhibited anti-gastric activity compared with the standard ranitidine, but this effect was not correlated with the presence of any compound. The same fact occurred with the MeOH extract of *P. alpina*, which showed anti-inflammatory abilities compared to the standard Indomethacin, and with the MeOH extract of *P. lapathifolium* var. *lanatum*, which offered analgesic activity at the same level of the standard aminopyrine. These results could be explained by the synergistic effects of the compounds present in the MeOH extracts instead of the specific action of any compound present in them. On the other hand, *α*-Santalone (isolated from the MeOH extract of *P. pubescens*) demonstrated to be the responsible for the potent analgesic activity of the extract, while flavonoids and sesquiterpene lactones may be responsible for the anti-inflammatory effect of *P. jucundum*. Finally, quercetin-3-*O*-β-d-glucuronide isolated from *P. perfoliatum* suppressed ear edema and peritoneal permeability in mice showing higher inhibition percentage respect to aspirin.

### 3.4. Antinociceptive Activity

Antinociceptive activities of the *n*-Hex, EtOAc and MeOH extracts from *P. hydropiper* were tested by acetic-acid-induced writhing method in Swiss albino mice of either sex [81]. Ethyl acetate extract showed a moderate dose-dependent effect, with writhing inhibition of 54.95% at a dose of 500 mg/kg compared with the standard aminopyrine (inhibition of 73.62% at dose of 50 mg/kg). Additionally, Oany et al. [82] reported that crude EtOH extract of *P. hydropiper* leaves exhibited a good antinociceptive activity compared to the standard drug aspirin, displaying a reduction in the abdominal writhing’s number of 41.02% and 69.23%, respectively, at a dose of 400 mg/kg b.w. Petroleum ether and chloroform (Chl) extracts of *P. barbata* showed antinociceptive properties, with an inhibition of writhing response of 46.8% and 44.8%, respectively (at a dose of 400 mg/kg b.w.), compared to 62.2% for the positive control aminopyrine [35]. The authors concluded that the compounds responsible for this activity were of apolar nature, possibly sterols or terpenoids. With respect to *P. acuminata*, the EtOH extracts of the leaves and stems exhibited antinociceptive activities with a percentage of writhing inhibition of 53.57% and 50%, respectively, at a dose of 500 mg/kg, similar values compared to the standard diclofenac (57.7% at a dose of 25 mg/kg) [9]. The antinociceptive activity of the crude MeOH extract of *P. verticillatum* rhizomes was tested by Khan et al. [193]. They found that at dose-dependency and through an opioid system, the extract could significantly reduce (72%) the number of writhes induced by acetic acid. The extract contained a high content of saponins and alkaloids.

### 3.5. Anticancer, Antitumoral and Cytotoxic Activity

Anticancer and antitumoral activities displayed by *Persicaria* and *Polygonum* species were studied in over the world. Many of these species have inhibitory abilities against different cancerogenic cell lines (Table 5): Jurkat, HL60, THP-1, CCRF-CEM, K562, U-937, K562 and P338 (human leukaemia); HepG2, Huh-7, SMMC-7721, HCCLM3, Hep3B (hepatocellular carcinoma); CaCo-2, HCT116, HT-29, CT-26, RKO, Colo320 and SW620 (colon cancer); MCF-7, HTB-26, MDA-MB-231-pcDNA3, MDA-MB-435 and Bcap-37 (breast cancer); SiHa, HeLa and PC-3 (cervical carcinoma); PC3 and PC3 (prostate carcinoma); LL2, A549, 95D and NCI H460 (lung carcinoma); SNU-601 and SGC-7901 (gastric cancer); PANC-1 (pancreatic adenocarcinoma); H22 (murine H22 ascitic hepatoma); EAC (Ehrlich ascites carcinoma); EBV-EA (Epstein-Barr virus early antigen); J82 (bladder transitional carcinoma); HEK 293 (renal cancer); NU-1066 (laryngeal cancer); OVCAR-3 (ovarian cancer); U87MG and BT-325 (glioblastoma); CAL 27 (oral adenosquamous carcinoma); Smur180 (sarcoma); and NIH3T3 (fibroblast).

According to this information, flavonoids glucosides provide the antitumoral activities to *P. amphibia*, *P. aplexicaulis*, *P. decipiens*, *P. glabra*, *P. limbate* and *P. senegalensis* with IC_50_ values varying from 0.01 to 100 µM depending on the cellular type under study. Other compounds such as gallic acid, ellagic acid and steroids seem to be responsible for the antitumoral activities of *P. bistorta*, *P. chinensis*, *P. hydropiper* and *P. bellardii* but show lower IC_50_ than flavonoids. Some specific compounds such as amplexicaule A, vanicoside B, lapathoside A and polygonumin A isolated from *P. amplexicaulis*, *P. lapathifolia* and *P. minor*, respectively, showed promising anticancer activities with IC_50_ values comparable to standard drugs.

### 3.6. Antiviral Activity

Some *Persicaria* and *Polygonum* species have shown antiviral abilities against different types of viruses (Table 6): HHV-1 and HSV-1 (human herpes virus 1); EBV-EA (Epstein-Barr virus); H1N1 A/PR/8/34, H1N2 A/HK/8/68 and B/Lee/40 (influenza A y B); HIV-1_VB59_ and HIV-1UG070 (immunodeficiency human virus); VACV-WR (vaccinia virus strain Western Reserve); VV (vaccinia virus); DEN-2 (dengue virus 2); VSV (vesicular stomatitis virus); RSV (respiratory syncytial virus); MCMV (cytomegaly virus); and HBV (hepatitis B virus). It could be highlighting the inhibition of HIV-1 protease (56%) of polygonumins A isolated from *P. minor* and the HIV-1 reverse transcriptase inhibitory activity of viscoazulone isolated from *P. viscosum*.

### 3.7. Antiparasitic Activity

Extracts and isolated compounds from *Persicaria* and *Polygonum* species were tested against different parasites that cause human diseases such as *Plasmodium falciparum* (causal agent of malaria disease); *Trypanosoma cruzi* and *T. brucei* (causal agents of Chagas disease); *Leishmania amazonensis* and *L. tropica* (causal agents of leishmaniasis disease); and *Pheretima posthuma* and other parasitic earthworms (which cause helminth infections). Results of these investigations are show in Table 7.

Few antiparasitic studies have been conducted using these two plant genera, and most of them were carried out with extracts. It could be remarked the anti-trypanocide activities of two chalcones obtained from *P. decipiens*, which showed IC_50_ values compared to the standard drugs diminazene and pentamidine against *T. brucei* and *T. congolense*. In addition, saponins from the butanolic extract of *P. hydropiper* displayed anti-leishmanial actions similar to the standard drug piperazine citrate.

### 3.8. Anti-Diabetic Activity

The anti-diabetic ability of EtOH extract from *P. aviculare* leaves was tested by Cai et al. [163]. The results showed that the extract is a potent inhibitor of α-glucosidase levels, higher than the standard drug acarbose, with IC_50_ values of 21.42 and 176.79 μg/mL, respectively. Additionally, the compounds myricitrin, quercetin, polydatin and isoquercitrin (present in the phenolic portion of *P. aviculare* extracts) were primarily responsible for this activity, with IC_50_ values of 8.65, 15.17, 35.15 and 148 μg/mL, respectively [163]. Ethanol extract of *P. pulchra* leaves showed a good α-glucosidase (involved in Type 2 diabetes mellitus) inhibition with IC_50_ value of 22.67 mg/mL, respect to the positive control acarbose (7.77 g/mL) [134]. In addition, Oany et al. [82] tested antihyperglycemic activity of crude EtOH extracts from leaves and stems of *P. hydropiper* and found that for all the doses tested (50, 100, 200 and 400 mg/kg b.w.), leaves’s extract showed higher effectiveness than standard glibenclamide, diminishing blood glucose levels between 48.8 and 58.2% for the extract and 42.1% for glibenclamide (at dose 10 mg/kg b.w.). Furthermore, the extracts of the leaves and stems (principally *n*-Hex, Chl and MeOH extracts) displayed α-amylase inhibitory activities, with IC_50_ values between 1.03 and 3.517 mg/mL [75]. This enzyme can increase the blood sugar level, as hydrolyses (1,4)-α-d-glycosidic linkages in polysaccharides contain three or more (1,4)-α-linked D-glucose units. Another study demonstrated that the EtOH extract of *P. cognatum* strongly inhibited α-amylase activity (86.6%) and moderately inhibited α-glucosidase activity (41.05%) with respect to the standard acarbose (58.4% and 57.56%, respectively) [169].

Kubinova et al. [97] tested the anti-diabetic activity of isolated compounds from the MeOH extract of *P. lapathifolia* aerial parts by the inhibition of AChE, BuChE and α/β-glucosidase. At a dose of 100 μM, kaempferol showed the highest inhibition of AChE (60.4%) and BuChE (74.5%), compared to the standard galantamine (95.7 and 47.9%, respectively), while gallic acid was only effective on AChE (inhibition of 51.2%). With respect to the α-glucosidase inhibitory assay, all isolated flavonoids showed potent enzyme inhibitory activity (72.6–97.2%) and eightfold higher activity than standard acarbose (11.4%), while for the β-glucosidase inhibitory assay, only quercitrin and isoquercitrin inhibited the enzyme with higher efficacy than the standard quercetin (23.6, 23.4 and 16.7%, respectively). Rodrigues et al. [177] tested the anti-diabetic ability of the leaf and root extracts of *P. maritimun* on α-amylase, baker’s yeast (α-glucosidase) and rat’s α-glucosidases. Both the MeOH and DCM extracts showed IC_50_ values lower than the positive control (acarbose) on α-glucosidase, but the MeOH extract had the highest capacity to inhibit the baker’s yeast α-glucosidase, with an IC_50_ value significantly lower than that of acarbose (IC_50_ value of 19 and 29 µg/mL for roots and leaves; 3144 µg/mL for acarbose).

The anti-hyperglycaemic effect of aqueous extract of *P. orientalis* flowers was tested on streptozotocin (STZ)-induced diabetic mice [130]. At a dose of 200 mg/kg, it was observed the most potent results: the extract significantly decreased blood glucose levels (144 mg/mL, 21 days) and serum cholesterol (53.2 mg/dL) compared to the control (210.06 and 82.4 mg/dL, respectively) and increased glycogen content in liver (30.7 mg/g for treatment and 11.86 mg/g for diabetic control). The hydroalcoholic (50%) extract of *P. senegalensis* leaves exhibited a potent anti-diabetic activity, inhibiting 100% of the α-glucosidase activity at a concentration of 10 ug/mL [144].

### 3.9. Antipyretic Activity

The antipyretic activity of the aqueous root extract of *P. bistorta* was studied in albino rats with Brewer’s-yeast-induced fever [45]. At a concentration of 100 mg/kg and after 4 h of treatment, the extract exhibited similar pyrexia activities to that of the standard drug acetaminophen, with a decrease in the rectal temperature of 0.8 °C. Another study revealed that the MeOH root extract of *P. glabra* exhibited a significant dose-dependent antipyretic activity on albino rats submitted to the Brewer’s-yeast-induced hyperpyrexia method [68]. Alkaloids, flavonoids and phenolic compounds could be responsible for this activity. Finally, Akhter et al. [89] tested the antipyretic activity of MeOH, EtOH, Chl, petroleum ether and *n*-Hex extracts of *P. hydropiper* leaves on Albino Swiss mice of both sexes. At doses of 200 and 400 mg/kg b.w., the MeOH, EtOH and Chl extracts showed a good activity compared to the standard drug paracetamol (50 mg/kg b.w.) by reducing temperature up to 4–7%, approximately.

### 3.10. Hepatoprotective Activity

El-Toumy et al. [172] tested the hepatoprotective activity of the MeOH extract of *P. equisetiforme*’s aerial parts on CCl_4_-induced hepatic illness in Sprague-Dawley rats by measuring the levels of serum marker enzyme activities (alanine amino transferase-ALT and aspartate amino transferase-AST) and the oxidative stress mediator levels (NO, malondialdehyde-MDA, glutathione-GSH, glutathione peroxidase-GPx and superoxide dismutase-SOD). The results showed that when the rats were pre-treated with the extract, they exhibited normal levels of ALT and AST (64.86 and 45.16 IU/mL, respectively, at an extract dose of 200 mg/kg) compared to the control (64.86 and 44.22 IU/mL). The GSH, GPx and SOD levels decreased, while the NO and MDA levels increased in comparison to the control. On the other hand, Christapher et al. [119] assessed the hepatotoxicity ability of the MeOH extract obtained from the leaves of *P. minor* on CCl_4_ and paracetamol-induced hepatotoxicity in Sprague-Dawley rats. They found that this extract significantly decreased the levels of AST, ALT, ALP and TB in both models compared to the CCl_4_ and paracetamol controls. At dose-dependency, the MeOH extract of *P. glabra* roots reduced the marker levels of hepatic injury such as serum glutamate oxaloacetate transaminase (SGOT), serum glutamate pyruvate transaminase (SGPT), alkaline phosphatase (ALKP) and total bilirubin in CCl_4_ and paracetamol-induced rats [120].

The anti-fibrotic effects of the aqueous and organic extracts of *P. plebeium* (whole plant) on inflammatory liver disease in CCl_4_-induced rats were tested, and the authors found that the extracts reduced the ALT, AST and gamma-glutamyl transpeptidase (γGT) levels in CCl_4_-induced rats and restored the CCl_4_-induced tissue fibrosis [190]. In addition, the aqueous extract from the roots of *P. bistorta* exhibited significant hepatoprotective effect in rats with CCl_4_-induced liver damage: the CCl_4_ administration on hepatic cells showed hydronic degeneration, swelling, congestion in portal vessels and sinusoids and optically empty cytoplasmic areas and increased the AST, ALT and total bilirubin levels in rats (173 and 223 IU/L and 1.15 mg/dL, repsectively) compared to the control (67.7 and 48.2 IU/L and 0.53 mg/dL, respectively). However, the cells recoupment when they were treated with aqueous roots extract of *P. bistorta* at a concentration of 100 mg/kg (levels of AST, ALT and total bilirubin decreased at 88.5 and 98.3 IU/L and 0.75 mg/dL, respectively) [46].

The hepatoprotective effects of the MeOH, EtOH and aqueous extracts of *P. amplexicaulis* rhizomes on albino mice were tested [34]. After the administration of CCl_4_ to the mice’s, ALT, AST, ALP and plasma bilirubin levels increased, and body weight decreased compared to the control; these levels were recouped when administrated aqueous, MeOH and EtOH extracts at a dose of 200 mg/kg. ALT, AST, ALP and plasma bilirubin decreased and body weight increased by 16.3, 12.96 and 8.08%, respectively, compared to the control silymarin (10.88% at a dose of 100 mg/kg). Furthermore, the EtOH fruit extract of *P. orientalis* exhibited significant hepatoprotective effects against CCl_4_-induced acute liver injury (ALI) in rats [131]. At doses of 0.5 and 1 g/kg of extract, the levels of AST, ALT, ALP, NO, tumour necrosis factor-α (TNF-α), interleukin-1β (IL-1β) and interleukin-6 (IL-6) decreased, while the levels of SOD, GPx and GRd increased. Protocatechuic acid, taxifolin, and quercetin identified by HPLC in the extracts, might be mainly responsible for these effects. In addition, Fan et al. [132] tested the potential inhibitory ability of OATP1B1 (anion transporting polypeptide 1B1, associated with drug-induced liver injury) of flavonoids isolated from *P. orientalis*. The results demonstrated that the compounds isoorientin and orientin showed weak inhibitory effects on OATP1B1-mediated fluvastatin consumption in OATP1B1-HEK293 cells (27.72 and 23.3%, respectively). Nguyen et al. [135] reported that the EtOAc extract from the aerial parts of *P. pulchra* and its subfractions exhibited a potent hepatoprotective activity in CCl_4_-induced rats since it reduced ALT activity between 50 and 68%. Finally, Xu et al. [189] reported that the total flavonoids isolated from *P. perfoliatum* exhibited potential hepatoprotective effect in CCl_4_-exposed mice, decreasing liver functional enzymes (ALT and AST), TNF-α, interleukin 6 and heat shock protein 90 (Hsp90) and increasing intrasplenic integrin β1, 5′-nucleotidase and antigen KI-67 cells at doses of 60 and 120 mg/kg/day.

### 3.11. Neuropharmacological Activity

#### 3.11.1. Anti-Alzheimer’s

The potential of β-sitosterol isolated from *P. hydropiper* for the management of Alzheimer’s disease was tested [90]. Authors observed a significant decline in free radical’s load in the brain tissues of the β-sitosterol-treated animals, with IC_50_ values of 140, 120, and 280 μg/mL from DPPH, ABTS and H_2_O_2_ assays, respectively. In addition, they performed memory assessment and coordination tasks (SWM, Y-maze and balance beam tests) and found that β-sitosterol-treated animals had gradual improvement in working memory and in motor coordination [90]. Previously, Ayaz et al. [78] reported that essential oil from the leaves and flowers of this plant demonstrated a significantly acetylcholinesterase (AChE) inhibitory ability (inhibition of 87 and 79.66%; IC_50_ = 120 and 220 μg/mL, respectively) and a butyrylcholinesterase (BChE) inhibition of 82.66 and 77.5%; IC_50_ of 225 μg/mL, respectively. Caryophyllene oxide and decahydronaphthalene were identified as the major components of the leaves’ and flower’s essential oils, respectively. Ahmad et al. [113] tested the AChE inhibition ability of different extracts (MeOH, EtOH, aqueous, DCM and *n*-Hex) and the essential oil of the leaves, stem and root of *P. minor.* Aqueous and MeOH leaf extracts showed the higher AChE inhibitory activity, with IC_50_ values of 234 and 342.77 µg/mL, respectively; root extracts had the lowest AChE inhibitory activity (IC_50_ > 1000 µg/mL). The presence of terpenoids could explain AChE activity of MeOH and aqueous extracts, as it can readily cross the blood–brain barrier by its small size and lipophilicity [113]. In addition, a study revealed that the aqueous extract of this plant could reverse scopolamine-induced memory deficits in a Barnes maze model (at a dose of 100 mg/kg) and could inhibit AChE activity by 68% with an IC_50_ of 0.04 mg/mL [121]. Regarding *P. glabra*, the MeOH and EtOH extracts of its aerial parts exhibited neuroprotective activity against AlCl_3_-induced (aluminium chloride) toxicity in rats, improving learning and memory and decreasing AChE levels at similar values of the standard rivastigmine at a concentration of 400 mg/kg [69]. Another study stated that the MeOH extracts from roots and aerial parts of *P. maritimum* could be a good alternative for the treatment of neurodegenerative disorders such as Alzheimer’s disease due to the ability for the inhibition of AChE and tyrosinase (TYRO), with IC_50_ values of 0.17 and 0.27 mg/mL for AChE and 0.59 and 0.6 mg/mL for TYRO [178].

#### 3.11.2. Anti-Depressant and Sedative

The anti-depressant activity of the aqueous extract of *P. glabra* was tested by Nizar et al. [70]. The results showed that the extract, in a dose-dependent manner (50, 100 and 200 mg/kg), induced a significant decrease in the immobility time of mice during behavioural despair test (BDT) and tail suspension test (TST) and increased the hyperactivity scores in an L-dopa-induced hyperactivity test, with similar values to the reference Imipramine (15 mg/kg). The depressant activity of four sesquiterpenes (viscosumic acid, viscozulenic acid, viscoazucine and viscoazulone) and the flavonoid glycoside quercetin-3-*O*-(6-feruloyl)-β-d-galactopyranoside isolated from the aerial parts of *P. viscosum* was tested by open-field test [196]. Viscoazucine and viscoazulone were the most potent depressants, showing gradual decreases in the movement of mice (No. of movements at 0 min = 143.97 and 137.95 and at 240 min = 23.92 and 27.93, respectively), while the other compounds showed a moderate depressant activity.

*P. hydropiper* showed anti-depressant and sedative activity: Sharif et al. [91] tested the antidepressant activity of the MeOH, EtOH and Chl leaf extracts by open field test and swimming test (mice were forced to swim and scored immobility). At a dose of 150 mg/kg, the Chl and EtOH extracts significantly decreased movements and exerted immobile phase in mice similar to the standard Imipramine (10 mg/kg). Devarajan et al. [92] examined the depressor effect of extracts of *P. hydropiper* leaves in salt-induced hypertension in mice and found that both could reduce blood pressure and heart rate in a dose-dependent manner. Shahed-Al-Mahmud and Lima [93] revealed that the MeOH leaf extract presented a good sedative and anxiolytic activity (at a concentration between 50 and 500 mg/kg) compared to a positive control of diazepam (1 mg/kg) using different test types such as open field test and thiopental sodium-induced sleeping time test in mice (for sedative activity) and elevated-plus maze and light-dark box (for anxiolytic activity). Finally, the ion channel-blocking activity K + of G protein-activated inwardly (responsible for maintaining the resting membrane potential and cell excitability) of different extracts of *P. maculosa* were tested through the automated patch-clamp method [109]. They found that Chl extract and its HPLC eluate fractions, at a concentration of 0.1 mg/mL, showed a potent K + channel-inhibitory activity compared to the reference compound Propafenone (between 62 and 76% and 71 and 81%, respectively).

#### 3.11.3. Neuroprotective Activity

Won and Ma [164] tested the neuroprotective activity of the aqueous-MeOH extract of *P. aviculare* by glutamate-induced neurotoxicity assay in primary cultures of rat cortical cells assay. The results showed that at a concentration of 100 µg/mL, a good neuroprotective potency of 50.1% compared to the standards CNQX (59.2%) and MK-801 (70.8%) was observed. Additionally, the juglandin extracted from crude *P. aviculare* exhibited a good neuroprotective activity in mice with LPS-induced Parkinson’s disease, attenuating memory impairments, promoting the expression of synaptic markers (SYP, PSD-95 and SNAP-25), decreasing production of pro-inflammatory cytokines (IL-1β, TNF-α, IL-18 and COX-2) and blocking TLR4/NF-κB pathway [165]. In addition, the extract of *P. aviculare* was tested for its neuro-inflammatory properties, and it was found that the extract could decrease lethargy-like behaviour and the compounds corticosterone, serotonin, and catecholamines (fatigue-related) in the brain and inhibited the production protein TNF-α (tumour necrosis factor) [166]. Finally, the neuroprotector effect of orientin (pyrone glucoside extracted from *P. orientalis*) was tested in pheochromocytoma cell line (PC12) stimulated by H_2_O_2_ in mice [133]. The results demonstrated that orientin was not toxic for PC12 cells and could reduce the H_2_O_2_-induced viability of PC12 cells at higher concentrations than 40 µg/mL. Moreover, orientin decreased H_2_O_2_-induced phosphorylation of signaling proteins (MAPKs, AKT and Src) and inhibited ROS (causing neurodegenerative diseases) accumulation in cells [133].

### 3.12. Diuretic Activity

In a Lipschitz test, the petroleum ether, Chl and EtOAc extracts of *P. barbata*’s aerial parts exhibited dose-dependent diuretic activity, but EtOAc extract (at a dose of 400 mg/kg) showed the most significant effect after 2 h of administration (diuretic activity = 1.77) compared to the standard Furosemide (diuretic activity = 1.72) [35]. In addition, the EtOAc extract of *P. lapathifolium* var. *lanatum* demonstrated a moderate to good diuretic activity, with values of 1.422 and 1.87 at doses of 150 mg/kg and 300 mg/kg b.w., respectively [98]. Finally, the diuretic activity of α-Santhalone isolated from *P. pubescens* aerial parts was tested by the Lipschitz methods, and it was found that at a concentration of 40 mg/kg, this compound presented a good activity after the first hour of administration (diuretic activity = 1.24) compared to the standard Furosemide 3 mg/kg (diuretic activity = 1.81) [191].

### 3.13. Gastroprotective Activity

The anti-ulcer activity of the alcoholic and aqueous extracts of *P. barbata* were tested by Pylorus ligation models and ethanol-induced gastric mucosal injury in rats [38]. They found that both extracts, in a dose-dependent manner, could significantly reduce the number and index of ulcers and total acidity, as well as increasing the pH index. The compounds present in both extracts such as saponins, sterols, glycosides and alkaloids could explain this activity [38]. In addition, the aqueous extract from the leaves of *P. chinensis* showed a gastroprotective effect against ethanol-induced gastric mucosal injuries in Sprague-Dawley rats, as it reduced gastric lesions and malondialdehyde levels (MDA) and increased superoxide dismutase level (SOD) [57]. The anti-ulcer activity of the aqueous extract from the leaves of *P. minor* against ethanol-induced gastric ulcers in rats was tested [122]. The extract showed a significant anti-ulcer activity compared to the standard omeprazole (20 mL/kg): at dose-dependentcy (250 and 500 mL/kg), the extract increased pH and gastric mucous, suppressed areas of gastric ulcer formation (35.33–188.17 mm) and inhibited 78.25–95.92% of the gastric ulcer. In addition, Qader et al. [123] obtained five fractions (hexane:ethyl acetate 1:1 *v/v* (F1), ethyl acetate:methanol 1:1 *v/v* (F2), methanol:acetonitrile 1:1 *v/v* (F3), acetonitrile:distilled water 1:1 *v/v* (F4) and distilled water 1:1 *v/v* (F5)) from the EtOH extract of the leaves of *P. minor* and tested its gastroprotective activity using the ethanol induction method in rats. All the fractions exhibited gastroprotective activity, but F2 showed the best result at a dose-dependency (very similar to Omeprazole values), inhibiting 90% of ulcer lesions and increasing mucus content (120 mg/g), SOD, hexosamine and PGE2 synthesis levels in the stomach wall mucosa.

The gastroprotective effect of the hydro-alcoholic root extract of *P. bistorta* was tested by indomethacin-induced gastric ulcer in rats [47]. At a dose-dependency (500 and 1000 mg/kg), this extract significantly increased mucus, SOD and catalase levels and decreased the ulcer index and thiobarbituric acid (TBARS), with similar values shown by the standard drug ranitidine (20 mg/kg), compared to the ulcer control group. Ayaz et al. [74] tested the gastroprotective ability of the crude MeOH extract from *P. hydropiper* and its fractions by aspirin-induced ulcerogenesis in rats. At a dose dependency (100, 200 and 400 mg/kg), the extract exhibited a good gastroprotective activity compared to the standard ranitidine (50 mg/kg), as it could decrease gastric juice volume, free acidity, total acidity and pepsin levels, as well as increased gastric juice pH levels. Additionally, the essential oil obtained from the leaves and saponins (fractions) showed the highest urease inhibition (> 70%, IC_50_ = 90 and 98 µg/mL, respectively) compared to the control drug thiourea (urease inhibition > 80% and IC_50_ = 80 µg/mL) [74].

### 3.14. Other Activities

The acetone extract from the aerial parts of *P. maritimum* is considered an interesting anti-melanogenic agent, as it was demonstrated to inhibit tyrosinase and melanin production in B16 4A5 melanoma cells, with IC_50_ values of 64.1 and 77.7 μg/mL, respectively [176]. Myricitrin, catechin and monogalloyl-hexose isolates could be responsible for this activity. Another study conducted by George et al. [124] explored the immunomodulatory properties of aqueous extract of *P. minor*, and they showed that this extract displayed significant phagocytic index (K) at doses of 200 and 400 mg/kg b.w. (K = 0.045 and 0.062, respectively) compared with the standard levamisole (K = 0.060). Finally, Kimura et al. [155] tested the inhibitory activity of flavonol *O*-glycosides isolated from 80% MeOH extracts of *P. tinctorea* leaves against HMG-CoA reductase. The fraction eluted with 100% MeOH showed the most potent inhibitory activity (56.7%), while five compounds isolated from this fraction were effective dose-dependently, inhibiting HMG-CoA reductase activity by 50 to 67.6%.

## 4. Conclusions

This *Polygonum* and *Persicaria* genera revision revealed the great variety of chemical constituents present in these plants, highlighting the bioactive groups of sesquiterpenes, flavonoids and phenolic acids. Methanolic, ethanolic, hexanic, ethyl acetate and water extracts and their fractions and chemical phytoconstituents have demonstrated different pharmacological activities such as antifungal, antibacterial, antioxidant, anti-inflammatory, anticancer and neuropharmacological activities in many publications, which were updated and revised here. These findings revealed that the species of the *Persicaria* and *Polygonum* genera could be property developed as good candidates for clinical assays in the future, allowing for the expansion of knowledge for the treatment of new diseases. *P. glabra*, *P. hydropiper*, *P. minor*, *P. lapathifolia* and *P. chinensis* were the species that showed the highest number of medicinal properties. Native species of Asia and Europe are ones with the largest number of studies in the world, followed by African species. However, American species are poorly studied or do not present any study (e.g., *P. ferruginea*, *P. hydropiperoides*, *P. punctata* and *P. paraguayense*), which invites the investigation of these species in the future.

The wide range of pharmacological properties of *Polygonum* and *Persicaria* species may offer a new therapeutic promise to cure different diseases and health complications. So, an accelerated progress should be made through experimental research including robust clinical trials, in order to generate natural medicines that allow to counter the negative effects on human health.

## Figures and Tables

**Table 1 molecules-26-05956-t001:** Origin, distribution and pharmacological uses of *Persicaria* and *Polygonum* species.

Species	Origin/Distribution	Pharmacological Activity	Ref.
*Persicaria* genus			
*Persicaria acuminata* (Kunth) M.Gómez	From Mexico to South America	Antifungal; Antinociceptive; Anti-malarial	[9,25,26]
*Persicaria alpina* (All.) H.Gross	Native to Europe and temperate Asia	Anti-inflammatory; Anti-helminthic	[27]
*Persicaria amphibia* (L.) Delarbre	Native to Europe, Asia, North America, and parts of Africa	Antifungal; Antibacterial; Anti-cancer	[28,29]
*Persicaria amplexicaulis* (D.Don) Ronse Decr.	Native to China, the Himalayas and Pakistan	Antioxidant; Anti-cancer; Hepatoprotective	[30,31,32,33,34]
*Persicaria barbata* (L.) H.Hara	Native to Southeast Asia	Anti-inflammatory; Antinociceptive; Anti-cancer; Diuretic; Gastroprotective	[35,36,37,38]
*Persicaria bistorta* (L.) Samp	Native to Europe and Central and West Asia	Antioxidant; Anti-inflammatory; Anti-cancer; Antipyretic; Hepatoprotective; Gastroprotective	[39,40,41,42,43,44,45,46,47]
*Persicaria capitata* (Buch.Ham. ex D.Don) H.Gross	Native to China, India, Nepal, Malaysia, Thailand, Vietnam and Sri Lanka	Antibacterial; Anti-inflammatory	[48]
*Persicaria chinensis* (L.) H. Gross	Native to South Asian regions with sub-tropical and warm climate	Antifungal; Antibacterial; Antioxidant; Anti-inflammatory; Anti-cancer; Antiviral; Anti-helminthic; Gastroprotective	[49,50,51,52,53,54,55,56,57]
*Persicaria decipiens* (R.Br.) K.L.Wilson	Native to Australia	Anti-cancer	[58,59]
*Persicaria ferruginea* (Wedd.) Soják	Native to temperate climates of South America countries	Antifungal; Antibacterial; Antiviral; Anti-trypanocide	[8,26,60,61]
*Persicaria glabra* (Willd.) M.Gómez	Native to North America and Eurasia	Antibacterial; Antioxidant; Anti-inflammatory; Anti-cancer; Antiviral; Anti-malarial; Anti-leishmanial; Antipyretic; Hepatoprotective; Neuropharmacological	[62,63,64,65,66,67,68,69,70]
*Persicaria hydropiper* (L.) Delarbe	Distributed in the northern hemisphere	Antifungal; Antibacterial; Antioxidant; Anti-inflammatory; Antinociceptive; Anti-cancer; Anti-trypanocide; Anti-helminthic; Anti-diabetic; Antipyretic; Neuropharmacological; Gastroprotective	[71,72,73,74,75,76,77,78,79,80,81,82,83,84,85,86,87,88,89,90,91,92,93]
*Persicaria hydropiperoides* (Michx.) Small	Native to America, from Canada to Argentina and Chile (introduced in Europe)	Antifungal; Antibacterial; Anti-leishmanial	[19,73,94,95]
*Persicaria lapathifolia* (L.) Delarbre	Native to Europe and Asia (introduced in America)	Antifungal; Antibacterial; Antioxidant; Anti-inflammatory; Anti-cancer; Antiviral; Anti-helminthic; Anti-diabetic; Diuretic	[96,97,98,99,100,101]
*Persicaria limbata* (Meisn.) H.Hara	Distributed in the southwest of Africa, Egypt and tropical Asia	Anti-cancer	[102,103]
*Persicaria maculosa* Gray	Native to Europe and Asia (introduced in North America)	Antifungal; Antibacterial; Antioxidant; Neuropharmacological	[11,104,105,106,107,108,109]
*Persicaria minor* (Huds.) Opiz	Native to Europe (introduced in Australia and America)	Antibacterial; Antioxidant; Anti-inflammatory; Anti-cancer; Antiviral; Hepatoprotective; Neuropharmacological; Gastroprotective	[110,111,112,113,114,115,116,117,118,119,120,121,122,123,124]
*Persicaria orientalis* (L.) Spach	Native to India (naturalized in America)	Antioxidant; Anti-inflammatory; Anti-cancer; Anti-diabetic; Hepatoprotective; Neuropharmacological	[125,126,127,128,129,130,131,132,133]
*Persicaria pulchra* (Blume) Soják	Distributed in India and Africa	Hepatoprotective	[134,135]
*Persicaria punctata* (Elliott) Small	Native to America (from Canada to Argentina and Chile)	Antifungal; Antibacterial; Anti-inflammatory; Antiviral	[136,137,138,139,140,141]
*Persicaria sagittata* (L.) H.Gross	Distributed in southeastern North America	Antioxidant	[142]
*Persicaria senegalensis* (Meisn.) Soják	Distributed in south-central Africa	Antifungal; Antibacterial; Antioxidant; Anti-cancer; Anti-diabetic	[58,108,143,144]
*Persicaria stagnina* (Buch.Ham. ex Meisn.) Qaiser	Distributed in Pakistan, India, Bangladesh and Myanmar	Anti-inflammatory; Anti-cancer	[37,145]
*Persicaria tinctoria* (Aiton) H.Gross	Native to Southeast Asia	Antibacterial; Antioxidant; Anti-inflammatory; Anti-cancer; Antiviral	[146,147,148,149,150,151,152,153,154,155]
*Persicaria vivipara* (L.) Ronse Decr.	Distributed in the High Arctic	Anti-inflammatory	[156]
*Polygonum* genus			
*Polygonum arenastrum* Boreau	Distributed in Europe, North Africa, Southwest Asia and North America	Antifungal	[157]
*Polygonum aviculare* L.	Native to Europe; It is distributed all over the world	Antifungal; Antibacterial; Antioxidant; Anti-cancer; Anti-diabetic; Neuropharmacological	[158,159,160,161,162,163,164,165,166]
*Polygonum bellardii* All.	Distributed in central-northern Europe, North Africa and Southwest Asia	Antifungal; Antibacterial; Antioxidant; Anti-inflammatory; Anti-cancer	[167,168]
*Polygonum cognatum* Meisn.	Distributed from central Asia to occidental Asia	Antifungal; Antibacterial; Antioxidant; Anti-cancer; Anti-diabetic	[169,170]
*Polygonum equisetiforme* Sm.	Distributed in southern Europe, North Africa and Southwest Asia	Antioxidant; Hepatoprotective	[171,172]
*Polygonum jucundum* Meisn.	Distributed in the Chinese provinces	Anti-inflammatory	[173]
*Polygonum maritimum* L.	Native to Europe. It is distributed in North Africa, Southwest Asia and North America	Antifungal; Antibacterial; Antioxidant; Anti-inflammatory; Anti-diabetic; Neuropharmacological	[174,175,176,177,178]
*Polygonum muricatum* Meisn.	Distributed in India, Malaysia and Nepal	Anti-helminthic	[179]
*Polygonum paleaceum* Wall.	Distributed in China and India	Antioxidant; Anti-inflammatory	[24,180]
*Polygonum perfoliatum* L.	Native to East Asia (China, Japan, Indonesia, Malaysia, Nepal, Korea and Philippines)	Antibacterial; Anti-inflammatory; Anti-cancer; Antiviral; Hepatoprotective	[20,181,182,183,184,185,186,187,188,189]
*Polygonum plebeium* R.Br.	Native to Madagascar, South Asia and New Zealand (introduced in United States and Australia)	Hepatoprotective	[190]
*Polygonum pubescens* Blume	Native to central-south Asia	Anti-inflammatory; Diuretic	[191]
*Polygonum thunbergii* Siebold & Zucc.	Native to Southeast Asia (China, India, Japan, Korea and Taiwan)	Anti-cancer	[192]
*Polygonum verticillatum* Biroli ex Colla		Antinociceptive	[193]
*Polygonum viscosum* Buch.Ham. ex D. Don	Native to Nepal and widely distributed in Bangladesh, northeast India, Japan and China	Anti-cancer; Antiviral; Anti-helminthic; Neuropharmacological	[194,195,196]

**Table 2 molecules-26-05956-t002:** Antimicrobial activity of *Persicaria* and *Polygonum* species against human fungal and bacterial pathogens. Extracts, compounds, part of plant used, antimicrobial activity, standard drug and references are shown. Extracts: DCM (dichloromethane); EtOH (ethanol); MeOH (methanol); Chl (chloroform); Hex (hexane); EtOAc (ethyl acetate); But (butanol). Part used: L (leaves); F (flowers); R (roots); Sp (sprouts); S (seeds); St (stems); WP (whole plant); AP (aerial parts).

Species	Extract/Isolated Compounds	Part Use	Pathogen	Growth Inhibition (mm or %)/MIC/IC_50_	Standard Drug	Ref.
*P. acuminata*	Polygodial (isolated from DCM extract)	L	*Candida albicans* and *Cryptococcus neoformans*	MIC 3.9–62.5 μg/mL	Amphotericin B (MIC 0.25–0.78 μg/mL)	[25]
	Polygodial, isopolygodial and drimenol (isolated from DCM extract)		*Microsporum gypseum, Trichophyton rubrum* and *T. mentagrophytes*	MIC 62.5 μg/mL	Amphotericin B (MIC 0.075–0.12 μg/mL)	
*P. amphibia*	Aqueous and EtOH	F, L	*Staphylococcus aureus*	11–14 mm	Chloramphenicol (inhibition of 27 mm)	[28]
*P. capitata*	Aqueous and EtOH (tannin-enriched and flavonoid-enriched fractions)	WP	*S. aureus*, *Escherichia coli*, *Neisseria gonorrhoeae*, *Klebsiella pneumoniae* and *Proteus mirabilis*	MIC 0.0022–1.37 mg/mL (Aqueous extract); 0.375–15 mg/mL (other fractions)	Ciprofloxacin (MIC 0.125–0.625 μg/mL)	[48]
	Aqueous and EtOH (gallic acid, triterpenoid and steroid-enriched fractions)		*N. gonorrhoeae*	MIC 0.375–4 mg/mL		
*P. chinensis*	MeOH, aqueous, Chl and petroleum ether extract	L	*C. albicans* and *C. krusei*	7–18.67 mm	Fluconazole, 30µg/disc (19.67–20.33 mm)	[49]
			*K. pneumoniae*, *Bacillus cereus*, *Streptococcus viridians*, *Corynebacterium diphtheriae*, *Enterobacter aerogenes. Pseudomonas aeruginosa* and *Corynebacterium diphtheriae*	7.33–22.33 mm	Amoxicillin, 10 µg/disc (13–30.67 mm)	
	MeOH, aqueous, Chl and petroleum ether extract	WP	*C. albicans*, *T. rubrum*, *Aspergillus niger*, *A. flavus* and *Cryptococcus neoformans*	11–21 mm (MIC 250–500 µg/mL)	Amphotericin B (22–28 mm, MIC 30 µg/mL)	[50]
			*K. pneumoniae*, *P. aeruginosa*, *Bacillus coagulans*, *B. subtilis*, *B. megaterium*, *B. aerogenes*, *Lactobacillus leichmanii* and *Salmonella typhi*	12–15 mm (MIC 250–500 µg/mL)	Tetracycline (25–38 mm, MIC 15.5–31.5 µg/mL)	
*P. ferruginea*	Cardamonin (isolated from DCM extract)	AP	*Epidermophyton floccosum*	MIC 6.2 µg/mL	Amphotericin B (MIC 0.4–0.75 µg/mL); Terbinafine (MIC 0.004–0.04 µg/mL)	[8]
	Crude MeOH extract, sub-extracts (*n*-Hex and DCM) and Pashanone isolated from DCM extract		*E. floccosum*, *M. gypseum*, *T. mentagrophytes* and *T. rubrum*	MIC 25–125 µg/mL		
	Pashanone (isolated from DCM)		*C. albicans*, *C. neoformans* and *Saccharomyces cerevisiae*	MIC 25–50 µg/mL		
*P. glabra*	MeOH and EtOAc extracts	WP	*B. subtilis* and *Proteus vulgaris*	4–7 mm (MIC of 0.5–1 mg/mL)		[62]
	2-methoxy-5-oxo-2,5-dihydrofuran-3-yl (2*E*)-(−)-3-phenylprop-2-enoate, 3-hydroxy-5-methoxystilbene and (-)-pinocembrin (isolated from MeOH extract)	AP	*Mycobacterium tuberculosis*	IC_50_ values of 2.27, 3.33 and 1.21 μg/mL, respectively		[63]
*P. hydropiper*	Confertifolin (isolated from essential oils)	L	*E. floccosum* and *Curvularia lunata*	MIC 7.81 μg/mL	For fungi: Ketoconazole (MIC < 12.5 μg/mL); for bacteria: Streptomycin (MIC 25 μg/mL)	[71]
			*Enterococcus faecalis*	MIC 31.25 μg/mL		
	Drimenol (isolated from essential oils)		*T. mentagrophytes*, *T. rubrum, T. simii* and *A. niger*	MIC < 12.5 μg/mL		[72]
	Chl extract	R	*A. niger*, *A. flavus*, *A. fumigatus* and *T. rubrum*	17–20 mm		
			*B. subtilis*, *B. megaterium*, *S. aureus*, *E. aerogenes*, *E. coli*, *P. aeruginosa*, *S. typhi* and *Shigella sonnei*	22–25 mm (MIC 16–64 µg/mL)	Kanamycin, 30 µg/disc (32.7–35 mm, MIC 2–8 µg/mL)	
	Polygodial	Sp	*C. albicans*, *C. krusei*, *C. neoformans*, *S. cerevisiae*, *T. mentagrophytes*, *T. rubrum* and *Penicillium marneffei*	MIC 0.39–6.25 µg/mL	Amphotericin B (MICs 0.2–1.56 µg/mL).	[73]
	Crude MeOH extract and its fractions (saponins, chloroform and ethyl acetate sub-extract)	WP	*Proteus mirabilis*	20–30 mm (MIC 25–40.5 μg/mL)	Ceftriaxone (35 mm, MIC 10 μg/mL)	[74]
	Acetone and EtOH extracts	L, St	*K. pneumoniae*, *Haemophilus influenzae*, *Morganella morganii*	10–19 mm	Ampicillin 10 µg (17–20 mm)	[75]
*P. hydropiperoides*	Polygodial (isolated from MeOH extract)	F, Sp	*C. albicans*, *C. krusei*, *C. neoformans*, *C. utilis*, *S. cerevisiae*, *T. mentagrophytes*, *T. rubrum*, *P. marneffei* and *P. chrysogenum*	MIC 0.78–12.5 µg/mL	Amphotericin B (MIC 0.2–1.56 µg/mL)	[73]
	EtOH extract	L	*S. aureus*	9 mm		[94]
	Triterpenoids, tannins and flavonoids (isolated from MeOH extract)	F, L	*Salmonella typhimurium*	16–19 mm	Thymol	[19]
*P. lapathifolia*	Pinostrobin chalcone and Pashanone (isolated from crude extract)	S	*Trichoderma* sp., *Fusarium* sp., *Aspergillus* sp. and *Penicillium* sp.	10–22 mm	Clotrimazole (15–23 mm)	[96]
			*E. coli*	12–18 mm	Gentamycin (17–19 mm)	
	Flavokawin and Pashanone (isolated from crude extract)		*S. aureus*	9–13 mm		
*P. maculosa*	DCM extract and isolated compounds (polygodial, isopolygodial and pinostrobin)	AP	*M. gypseum*, *T. rubrum* and *T. mentagrophytes*	MIC 7.8–62.5 μg/mL	Ketoconazole (MIC 0.02–0.25 μg/mL) and Amphotericin (MIC 0.07–0.5 μg/mL)	[11]
	Polygodial (isolated from DCM extract		*C. albicans*, *C. neoformans* and *S. cerevisiae*,	MIC 15.6–500 μg/mL		
	Persilben	WP	*Trychophyton* sp.	MIC 125–250 mg/L		[104]
	MeOH extract	L	*E. coli*	100%	Ampicillin, 1 mg/mL	[105]
			*S. typhi* and *P. aeruginosa*	42–49%		
	Quinic, gallic and chlorogenic acid and quercetin 3-*O*-β-d-glucopyranoside (isolated from EtOH extract)	AP	*P. aeruginosa* and *Salmonella enterica*	Inhibition of biofilm formation of 50%	Dimethyl sulfoxide 0.1% (DMSO)	[106]
			Pyocyanin production (toxin secreted by *P. aeruginosa)*	47%		
*P. minor*	MeOH and EtOH extracts (50 and 70%)	L	*E. coli*, *B. subtilis* and *S. aureus*	11.9–16.2 mm		[110]
	MeOH, Chl and petroleum ether extracts		*Helicobacter pylori*	12.3–15.5 mm		[111]
	Aqueous-EtOH (30%) and aqueous (100%) extracts		*E, faecalis*, *E. coli* and *S. aureus*	16.45–19.5 mm (concentration of 200 mg/mL)	Penicillin (20.7–25.5 mm at concentration of 10 mg/mL)	[112]
	Hex, DCM and MeOH extracts		*B. cereus*	12.5–14.5 mm (MIC of 1.25–2.5 mg/mL)	Ampicillin (17.5 mm; MIC of 0.1 mg/mL)	[113]
*P. punctata*	Polygodial (isolated from DCM extract) and DCM extract	AP	*C. albicans*, *A. niger* and *Mucor sp.*			[136,137]
			*B. subtilis*, *S. aureus* and *Micrococcus luteus*			
	MeOH extract		*E. faecalis*, *S. aureus*, *B. subtilis* and *Mycobacterium phlei*			[138]
	Isotadeonal and ethyl ether extract		*P. aeruginosa* and *S. aureus*	75% (concentration of 100 µg/mL)		[139]
*P. senegalensis*	Pyrazolines derivates of chalcones	AP	*C. krusei*, *C. neoformans*, *S. aureus* and *C. glabrata*,	IC_50_ 7.56–13.74 µg/mL	Amphotericin B (IC_50_ 0.37–1.38 µg/mL) and Ciprofloxacin (IC_50_ 0.09 µg/mL)	[143]
	Hydroalcoholic (50%) extract	L	*E. faecalis*, *B. subtilis* and *S. aureus*	MIC 1.25–5 mg/mL		[144]
*P. tinctoria*	Extract and tryptanthrin isolated from this extract	L	*Streptococcus mutans*, *S. sobrinus**, Porphyromonas gingivalis, Campylobacter rectus, Prevotella intermedia* and *Actinobacillus actinomycetemcomitans*	MIC 1.74–3.48 µg/mL (for extract) and 6.25–25 ug/mL (for tryptanthrin)		[146]
	Kaempferol (isolated from leaves extract)		*S. mutans, S. sobrinus, P. gingivalis* and *P. intermedia*	MIC 25–50 µg/mL		
	Tryptanthrin	WP	*H. pylori*	Inhibited 100% colony formation (dose of 10 µg/mL)	Amoxicillin, clarithromycin and omeprazole	[147]
*P. arenastrum*	MeOH extract	L, St	*C. albicans*	MIC 250 μg/mL	Chloramphenicol (MIC 0.156–1.25 μg/mL), amphotericin B and ketoconazole (MIC 0.04–0.31 μg/mL)	[157]
			*C. krusei*	MIC 62.5–15.63 μg/mL		
*P. aviculare*	Chl extract	St	*A. niger, A. flavus* and *A. fumigatus*	14–18 mm (MIC 1–5 mg/mL)	Cotrimoxazole antibiotic, 10 mg/g (18–34 mm)	[158]
			*E. coli, P, mirabilis, P. aeruginosa, S. typhi, S. aureus* and *B. subtilis*	24–28 mm (MIC 8–15 mg/mL)		
	EtOH extract	AP	*P. aeruginosa, S. aureus* and *Acinetobacter baumannii*	74–100% (at a concentration 1 mg/mL)		[159]
*P. bellardii*	MeOH and EtOAc extracts, *n*-Hex, Chl and *n*-But fractions	AP	*C. albicans*	11–20 mm (MIC 1–25 mg/mL)	Nystatin antibiotics, 25 µg/disc	[167]
			*S. aureus, B. subtilis, E. coli* and *P. aeruginosa*	11–30 mm (MIC 1–5 mg/mL)	Cefotax, 15–30 ug/disc	
*P. cognatum*	EtOH extract	L	*C. albicans*	MIC 2.5 mg/mL		[169]
			*S. aureus, P. aeruginosa* and *E. coli*	MIC 0.156–0.625 mg/mL		
		WP	*K. pneumoniae, S. aureus, E. coli, B. megatarium, C. albicans*	8–10 mm	Ceftriaxone and Nystatin 30 µg/disk (10–11 mm)	[170]
*P. maritimum*	Phenolic compounds (isolated from EtOH extract)	AP	*Penicillium* sp., *Aspergillus* sp., *Alternaria alternata* and *Fusarium semitectum*	19–34% (concentration of 1–5 mg/mL)		[174]
	Crude extract		*B. cereus, B. subtilis, S. aureus, A. baumannii, E. faecalis, P. mirabilis* and *Citrobacter freundii*	MIC 0.12–4.02 mg/mL		[175]
			*E. coli* and *P. aeruginosa*	MIC 16.08–64.35 mg/mL		
*P. perfoliatum*	EtOAc fraction	AP	*S. aureus* and *Cutibacterium acnes*	MIC of 0.25%,	Quercetin (MIC of 0.06–0.3%) and methyl paraben (MIC 0.13–0.25%)	[181]
	Water extract		*S. aureus, E. coli, Streptococcus* sp., *Salmonella* sp. and *Pasteurella* sp.	0.56–21.86 mm (MIC 0.031–0.063 mg/mL)		[182]
	EtOH (75%)		*S. aureus, B. subtilis* and *P. aeruginosa*	MIC 5–10 mg/mL		[183]

**Table 3 molecules-26-05956-t003:** Antioxidant activity of *Persicaria* and *Polygonum* species. Extracts, compounds, part of plant used, antioxidant activity and references are shown. Extracts: DCM (dichloromethane); EtOH (ethanol); MeOH (methanol); EtOAc (ethyl acetate); But (butanol). Part used: L (leaves); F (flowers); R (roots); Sp (sprouts); Sh (shoots); S (seeds); St (stems); Rh (rhizomes); WP (whole plant); AP (aerial parts). Antioxidant assays: DPPH (2,2-diphenyl-2-picryl hydroxyl); TEAC (Trolox Equivalent Antioxidant Capacity); CUPRAC (Cupric Reducing Antioxidant Capacity); ABTS ((2,2′-azino-bis(3-ethylbenzothiazoline-6-sulfonic acid)); NBT (Nitroblue tetrazolium); FRAP (Ferric Reducing Antioxidant Power); ORAC (Oxygen Radical Absorbance Capacity assay); CCA (Copper Chelating Activity).

Species	Extract/Isolated Compounds	Part Used	Summarized Bioactivity	Ref.
*P. equisetiforme*	MeOH extract	Sh	DPPH scavenging activity from 12 to 51 mM TRE/g DW. EC_50_ of reducing power = 68–210 μg/mL. This capacity was attributed to different compounds such as quinic acid, gallic acid, (+ )-catechin, epicatechin, quercetin-3-*O*-β-d-galactoside, quercetin-3-*O*-α-l-rhamnoside and cirsiliol.	[171]
*P. glabra*	Flavonoids, phenols, tannins, terpenoids and reducing sugars (isolated from MeOH extract)	L	DPPH free radicals with inhibitory concentration (IC_50_) of 79.81 μg/mL.	[64]
*P. lapathifolia*	Isoquercitrin, hyperoside, quercitrin and taxifolin, gallic acid (isolated from MeOH extract)	AP	At a dose of 5 μM, compounds showed higher antioxidant activity than the standard quercetin (TEAC value of 1.16 μM for compounds and 1.1 μM for standard quercetin). Gallic acid was the most potent scavenger of hydroxyl radicals (inhibition of 70.8%) compared to the standards quercetin and Superoxide Dismutase (SOD) (inhibition of 66.2 and 77.2%, respectively).	[97]
*P. tinctoria*	Flavonol *O*-glycosides with TMF as an aglycone	AP, S, Sp	DPPH scavenging activity from 500 µmol/g DW, (AP), 100 µmol/g DW (Sp) and 50 µmol/g DW (S).	[148]
	EtOAc fraction	St	Displayed high antioxidant activity (IC₅₀ 7.17 µg/mL) with respect to L-ascorbic acid (IC₅₀ 5.5 µg/mL).	[149]
	MeOH extract	L, F	ABTS radical scavenging of 99.12 and 96.35 M TE/g DW (L and F, respectively). CUPRAC values of 78.37 and 86.22 M TE/g DW (L and F, respectively).	[150]
*P. aviculare*	Lyophilized EtOH extract	WP	DPPH and FRAP inhibition greater than 75% at a concentration of 50 µg/mL. NBT greater than 90% at a concentration of 5 µg/mL. Extract showed it can protect DNA in hydroxyl-radical-induced DNA strand scission assays.	[160]
*P. cognatum*	EtOH extract	L	DPPH radical scavenging maximum of 18% and ABTS radical scavenging of 70%. Positive control BHT (DPPH and ABTS scavenging of 80%).	[169]
*P. maculosa*	EtOH extract	AP	SC_50_ (concentration that scavenges the free radicals by 50%) of 12.5 µg/mL. FRAP value of 1.6 mmol TE/g extract.	[107]
	Persilben	AP	A 2.7 µM solution of persilben induced neutralization of DPPH radical by 40% (increase in content of persilben did not cause further reduction of DPPH).	[104]
	MeOH extract	AP	DPPH = 93.02% and FRAP = 7.3 mg/g. Standard BHT = 86.5%.	[108]
*P. senegalensis*	MeOH extract	AP	DPPH = 68.13% and FRAP = 6.2 mg/g. Standard BHT = 86.5%.	[108]
	Hydroalcoholic (50%) extract	L	IC_50_ value for DPPH radical scavenging activities = 6.8 μg/mL. Positive control: L-ascorbic acid (IC_50_ value 1.25 μg/mL).	[144]
*P. bistorta*	MeOH and EtOH extracts	R	IC_50_ value of 49.20 µg/mL (MeOH) and 61.14 µg/mL (EtOH).	[39]
	MeOH extract and zinc oxide nanoparticles (ZnO-NPs)	R	MeOH extract showed higher DPPH and diammonium salt radical scavenging activity. Moreover, ZnO-NPs synthesized from root can inhibit the ABTS radicals, with IC_50_ value of 40 µg/mL and a dose-dependent activity.	[40]
*P. amplexicaulis*	Nine known compounds and a previously undetermined one (5, 6-dihydropyranobenzopyrone), amplexicine and gallic acid (isolated from EtOH extract)	AP	IC_50_ values of 10.2 µmol/L (5, 6-dihydropyranobenzopyrone), 12.2 µmol/L (amplexicine) and 14.4 µmol/L (gallic acid).	[30]
	Crude MeOH extract and their fractions (But, EtOH, EtOAc and aqueous)	Sh, L, Rh	All fractions and parts of plant displayed antioxidant activity (IC_50_ between 1.03 and 58.2 μg/mL), but leaf crude MeOH and EtOAc fraction were the most effective for radical scavenging activity DPPH, with IC_50_ = 1.03 and 3.1µg/mL, respectively.	[31]
*P. chinensis*	MeOH extract	WP, St, L	Potent antioxidant activity respect to the standard L-ascorbic acid and Rutin: IC_50_ from DPPH of 7.03–19.13 µg/mL. IC_50_ from lipid peroxidation of 16.32–25.31 µg/mL. IC_50_ from hydrogen peroxide method of 28.12–60.01 µg/mL.	[50]
	EtOAc fraction	AP	ORAC value of 0.965 μmol Trolox/mg.	[51]
*P. chinense* var. *chinense* and *P. chinense* var. *hispidum*	Aqueous extract		Exhibited moderate antioxidant activity, with IC_50_ values from 180.87 to 255.69 μg/mL (from *chinense* variety) and 182.96 to 250.84 μg/mL (from *hispidum* variety). Gallic acid, chlorogenic acid, ellagic acid, quercitrin and brevifolin carboxylic acid isolated from these species could explain the antioxidant ability.	[52]
*P. minor*	MeOH extract	L	DPPH inhibition of 80.3% and FRAP value of 377 µMol Fe (II) g^−1^.	[110]
	Aqueous and EtOH extracts	L	DPPH radical scavenging of 81.88 and 89.5% and FRAP value of 849.33 and 11,220 mmol/g, from aqueous and EtOH extracts, respectively. Authors have hypothesized that high levels of phenolic compounds (TPC 55.5–207 mg/g) may be the reason for the high antioxidant activity of this species.	[114]
*P. bellardii*	EtOAc extract and isolated compounds	AP	EtOAc extract showed at DPPH inhibition between 29.9 to 82.5%. Gallic acid and quercetin showed a minimum inhibition of 42.0–43.1% and maximum inhibition of 97.1–100%, values closely related to the reference’s L-ascorbic acid and quercetin (45–99.6%).	[167]
*P. hydropiper*	EtOAc fraction	L	Strongly inhibited free radicals with an IC_50_ value of 13.3 mg/mL.	[76]
	Flavonoids belonging to the quercetin family	L	TEAC values of 3.46–6.14.	[77]
	Essential oil	L, F	Significantly DPPH free radical scavenging, ABTS and H_2_O_2_, with IC_50_ values of 20, 180 and 45 μg/mL (for essential oil obtained from leaves) and 200, 60 and 50 μg/mL (for essential oil obtained from flowers), respectively.	[78]
*P. paleaceum*	Crude extract, EtOAc and But fractions	Rh	A good capacity on DPPH, with SC_50_ values of 16.72 µg/mL (crude extract), 10.64 µg/mL (EtOAc fraction) and 30.65 µg/mL (But fraction). Gallic acid, caffeic acid derivatives and procyanidin are the main compounds that play an important role for the antioxidant activity of this plant.	[24]
*P. sagittata*	Gallic acid, methyl gallate, vanicoside A, quercetin, protocatechuic acid and vanicoside B (isolated from acetone extract)	St	Gallic acid showed the most potent DPPH scavenging activity (IC_50_ 8.88 µM), followed by methyl gallate, vanicoside A, quercetin, protocatechuic acid and vanicoside B, with IC_50_ values of 15.37, 26.82, 29.18, 32.38 and 35.06 µM, respectively. Positive control: L-ascorbic acid (IC_50_ = 30.49 µM).	[142]
*P. maritimum*	Acetone extract	AP	High O_2_ radical dot scavenging (RSA of O_2_) (IC_50_ = 40.4 μg/mL) and a moderate total antioxidant capacity and anti-lipid peroxidation (IC_50_ of 647 and 784 μg/mL).	[176]
	MeOH extract	L	IC_50_ from DPPH of 26 µg/mL, IC_50_ from FRAP of 48 µg/mL and IC_50_ from CCA of 770 µg/mL. Benzoic acid, phloroglucinol, phytol and linolenic acid were identified as possible compounds responsible for these bioactivities.	[177]
	Crude extracts	AP	Good antioxidant scavenging effects on DPPH radical (7.71 µg/mL). Positive controls: BHA, L-ascorbic acid and quercetin (DPPH 2.59–2.61 µg/mL).	[175]
*P. orientalis*	Taxifolin	L, S	Inhibitory effect on DPPH radical of 100% at a concentration of 7.5 µmol/L and the IC_50_ value for taxifolin was 4.11 mmol/L.	[125]

**Table 4 molecules-26-05956-t004:** Analgesic an anti-inflammatory activity of *Persicaria* and *Polygonum* species. Extracts, compounds, part of plant used, anti-inflammatory and analgesic activity with their references are shown. Extracts: DCM (dichloromethane); EtOH (ethanol); MeOH (methanol); Hex (hexane); EtOAc (ethyl acetate); But (butanol). Part used: L (leaves); St (stems); Rh (rhizomes); Sta (stalks); WP (whole plant); AP (aerial parts).

Species	Extract/Isolated Compounds	Part Used	Summarized Bioactivity	Ref.
*P. chinensis*	MeOH extract	AP	At a concentration of 300 μg/mL, extract significantly inhibited regulation of nitric oxide (NO) at 72% in RAW264.7 cells and prostaglandin (PGE2) production was strongly suppressed up to 53%. These authors tested a murine HCl/EtOH-induced gastric ulcer model to evaluate the anti-inflammatory activity in vivo and found that the extract exhibited a significant anti-gastric activity, compared with the standard anti-ulcer ranitidine (40 mg/kg).	[53]
*P. chinensis* var. *hispidum*	Aqueous extract	AP	Ellagic acid and quercitrin inhibited the development of xylene-induced ear edema, with significant inhibition at a dose of 400 mg/kg.	[52]
*P. alpina*	MeOH extract	Rh	HRBC (Human Red Blood Cell) membrane stabilization method and percentage of inhibition protein denaturation method were used for tested in vitro anti-inflammatory activity of MeOH and aqueous extracts which showed a good anti-inflammatory ability, with a membrane stabilizing activity of 81.29% and an inhibition of protein denaturation of 72.70%, compared to the standard Indomethacin (95.56 and 88.26%, respectively).	[27]
*P. hydropiper*	MeOH extract	L	Extract blocked the production of NO, PGE2 and tumour necrosis factor on RAW264.7 cells and peritoneal macrophages.	[79]
	Aqueous extract	Sta	Extract attenuated the weight and length ratio of the colon, ameliorated the activity of MPO and the GSH content and regulated Cox-2, TNF-α and IL-1β levels in rats with TNBS-induced intestinal inflammation.	[80]
*P. pubescens*	*α*-Santalone (isolated from MeOH extract)	AP	The compound showed the most potent analgesic activity at a dose of 40 mg/kg b.w. (body weight), with an inhibition of acetic-acid-induced writhing response of 48.9%, compared to 62.2% for standard aminopyrine.	[191]
*P. lapathifolium* var. *lanatum*	Hex and MeOH extracts	WP	Anti-inflammatory activity: extracts inhibited carrageenan induced inflammation in rat paw at 41.09% (Hex) and 30.15% (MeOH), with a dose of 300 mg/kg b.w. Standard drug: phenylbutazone (42.15% inhibition; at a dose of 100 mg/kg b.w.). Analgesic activity: MeOH extract showed the highest inhibition of acetic acid-induced writhing reflex (62.29%; dose of 300 mg/kg b.w.), Standard drug: aminopyrine (69.94%).	[98]
*P. bellardi*	MeOH extract, their fractions (EtOAc and But) and isolated compounds (quercetin/its derivatives and myricetin/its derivatives)	AP	But extract was the most inhibitor of 5-lipoxygenase (5-LOX) (IC_50_ 14.20 mg/mL), followed by EtOAc extract, MeOH extract, myricetin and quercetin (23.16, 24, 34.25 and 43.81 mg/mL, respectively). In addition, EtOAc and But extracts, myricetin and its glycosylated derivatives showed a significant inhibition of PGE2 release (15.23–42.81%).	[168]
*P. orientalis*	MeOH extract	L	At dose-dependent, extract exhibited a moderate inhibition percentage of haemolysis (50.37%). Positive control: hydrocortisone (inhibition of 86.56%). Extract showed a good percentage of inhibition protein denaturation (79.22%). Positive control: diclofenac Na (inhibition of 86.85%).	[126]
	EtOAc and ethyl ether extracts	St, L	All doses tested (3.75, 5 and 7.5 g/kg) of the two extracts showed anti-inflammatory and analgesic activity, significantly inhibiting ear edema and significantly decreasing writhing in mice.	[127]
*P. stagnina*	Hex, EtOAc and MeOH extracts	AP	The most potent analgesic activity was observed with the EtOAc extract (writhing inhibition of 50.3% at a dose of 400 mg/kg b.w.), while Hex extract showed the highest levels of anti-inflammatory activity (carrageenan-induced edema inhibition of 60.1% at a dose of 200 mg/kg b.w.), a much better effect than that of the conventional anti-inflammatory agent phenylbutazone (maximum 38.3% after 4 h).	[145]
*P. maritimum*	MeOH and DCM extracts	L	Extracts showed significant inhibition of NO production by LPS-stimulated RAW 264.7, at a concentration of 100 µg/mL. *β*-sitosterol, stigmasterol, 1-octacosanol and linolenic acid were identified as the possible compounds responsible for these bioactivities.	[177]
	Acetone extract	AP	Extract showed a potent ability to reduce NO production on LPS-stimulated RAW 264.7 macrophages (IC_50_ of 22.0 μg/mL). Positive control: L-NAME (IC_50_ of 27.6 μg/mL).	[176]
*P. jucundum*	EtOH extract	AP	Extract inhibited inflammatory reactions that cause instant irritation of the mouse ear, significantly inhibiting inflammatory mediators such as RAW264.7 cells (amurine macrophage cell line), production of NO, tumour necrosis factor TNF-α and IL-6 production in a dose-dependent manner. Flavonoids and sesquiterpene lactones may be responsible for the anti-inflammatory effect.	[173]
*P. minor*	Aqueous and EtOH extracts	AP	At a dose of 30 µg/mL, EtOH extract inhibited the activities of lipoxygenase and cyclooxygenase-1, while the aqueous extract completely reduced paw edema induced by λ-carrageenan at doses of 100 or 300 mg/kg b.w.	[115]
*P. punctata*	Decoction and EtOH-water extract	WP	Exhibited anti-inflammatory activity against the carrageenan-induced pedal edema/Gastric intubation in vivo.	[140]
*P. tinctoria*	Tryptanthrin (isolated from EtOH extract)	L	Compound significantly inhibited the protein expression of iNOS and COX-2, suppressed the activation of p38 MAPK pathway and inhibited the TLR4 and MyD88 protein expression in LPS-stimulated BV2 microglial cells.	[151]
	Polyphenolic fraction	L	Reduced NO synthesis in murine RAW264 macrophage cells stimulated with LPS, which showed a good analgesic activity.	[148]
	Extracts	St	Exhibited protective effects of DNA damage against oxidative stress and anti-inflammatory effects by its capacity for NO suppression in LPS-induced RAW264.7 cells.	[149]
*P. barbata*	Petroleum ether extract	AP	Inhibition of 39.3% paw edema after 2 h at a dose of 400 mg/kg b.w. (inhibitory ability slightly higher than that of the conventional anti-inflammatory agent phenylbutazone, 38.3% after 4 h).	[35]
*P. vivipara*	2-propanol extract		Exhibited anti-inflammatory activity against LPS-induced inflammation in RAW264.7 macrophages (IC_50_ = 270 µg/mL) by inhibiting NO, prostaglandin, interleukin and tumour necrosis factor (TNF)-α release at similar levels as positive control.	[156]
*P. bistorta*	But extract		Extract exhibited analgesic effect, as it could reduce the writhing times of the mice induced by acetic acid and raised the threshold of pain induced by hot and electric stimulation.	[41]
*P. paleaceum*	Extracts		Extracts reduced malondialdehyde (MDA) content in inflamed paws, inhibited NO synthase and β-NAG activities and significantly reduced the content of NO, IL-1β and TNFα in exudates.	[180]
*P. glabra*	Quercetin and quercetin glycosides	L	Compounds showed maximum effect at 90 min with latency time of 18.78 and 15.07 min, respectively (at doses of 200 mg/kg); the chemically induced writhing tests (for the evaluation of peripheral analgesic activity) showed maximum inhibition of writhing = 74.18% (quercetin) and 61.73% (quercetin glycosides) at a dose of 200 mg/kg in comparison with aspirin (dose of 50 mg/kg; inhibition of writhing = 78.41%).	[65]
*P. capitata*	Aqueous and EtOH extracts	WP	Significant inhibition of edema in animal models (76.19% and 71.13%, respectively) at a dose of 0.30 g/kg.	[48]
*P. perfoliatum*	Quercetin-3-*O*-β-d-glucuronide		At a concentration of 8 mg/kg, compound suppressed ear edema induced by dimethyl benzene and peritoneal permeability induced by acetic acid in mice (45.96 and 40.10%, respectively), showing higher inhibition percentage respect to aspirin (24.62 and 34.38%, concentration 100 mg/kg).	[184]

**Table 5 molecules-26-05956-t005:** Anticancer and antitumoral activities of *Persicaria* and *Polygonum* species. Extracts, compounds, part of plant used, cell line and references are shown. Extracts: EtOH (ethanol); MeOH (methanol); Chl (chloroform); Hex (hexane); EtOAc (ethyl acetate); But (butanol). Part used: L (leaves); F (flowers); Fr (fruits); R (roots); Sp (sprouts); S (seeds); St (stems); Rz (rhizomes); T (tubers); B (barks); WP (whole plant); AP (aerial parts).

Species	Extract/Isolated Compounds	Part Use	Cell Line	Summarized Bioactivity	Ref.
*P. amphibia*	Flavonoid glucosides: quercetin 3-*O*-β-d-glucopyranoside and quercetin-3-*O*-α-rhamnosyl-(1-2)-β-glucoside (isolated from butanoic fraction)	AP	Jurkat	EC_50_ 1.2 and 0.12 µM.	[29]
			HL60	EC_50_ 0.98 and 0.01 µM.	
*P. amplexicaulis*	Flavonoids (isolated from EtOH extract)	Rz	HepG2, Huh-7, H22, SMMC-7721	At a dose-time dependence: reduced cell viability, induced cell apoptosis and increased expression of SHP-1 (tyrosine phosphatase catalysing STAT3 dephosphorylation protein).	[32]
	Amplexicaule A (isolated from EtOH extract)		MCF-7 and MDA-MB-435	At a concentration of 150 mg/kg: suppressed tumour mass in 0.6–0.7 g (respect to positive control Capecitabine 10 mg/kg, 0.5–0.6 g), induced apoptosis in cancer cells, increased caspase-3, -8, -9 and PARP (enzymes and proteins that catalyse apoptosis) levels and suppressed MCL-1 and BCL-2 expression.	[33]
*P. barbata*	Methyl (2*S*,3*S*)-2-(3,4-dimethoxyphenyl)-4-((*E*)-3-ethoxy-3-oxoprop-1-en-1-yl)-7-methoxy-2,3 dihydrobenzo-furan-3-carboxylate (1) and (*E*)-3-((2*S*,3*S*)-2-(3,4-dimethoxyphenyl)-7-methoxy-3-(methoxy carbonyl)-2,3-dihydrobenzofuran-4-yl) acrylic acid (2) (isolated from EtOAc fraction)	WP	CAL-27 and NCI H460	IC_50_ of 48.52 and 53.24 µM (for 1) and 86.95 and 93.34 µM (for 2). Standard drugs: 5-Fluorouracil, IC_50_ 97.76 µM (for CAL-27) and Cisplatin, IC_50_ 19 µM (for NCI H460). Compound **1** induced apoptosis in CAL-27 cell line after 24 to 48 h treatment.	[36]
	Petroleum ether extract	AP	Potato disc assay	Moderate activity, with inhibition of tumour growth of 57.1% at a concentration of 400 µg/disc and IC_50_ value of 290 µg/disc.Positive control: vincristine sulphate 3.125 µg/disc, 100% inhibition tumour).	[37]
*P. bistorta*	Gallic acid, protocatechuic acid, syringic acid, catechol, syringol, 4-methyl catechol, myristic acid (isolated from MeOH-water extracts)	Rz	HCCLM3	Cell viability < 30% (at a dose of 200 µg/mL) and GI_50_ values between 86.5 to 126.8 µg/mL.	[42]
	Aqueous extract	WP	Hep3B	Inhibited autophagosome and proteasome activity, resulting in restriction of cell motility and apoptosis induction in Hep3B cells.	[43]
	Zinc oxide nanoparticles (ZnO-NPs) (synthesized using *P. bistorta* extract)	R	MCF-7	Effective dose-dependent activity, with a percentage of cells viability < 10% (at a concentration of 125 µg/mL).	[40]
	*n*-Hex and Chl fractions and its sub-fractions	Rz	P338, HepG2, J82, HL60, MCF7, LL2	Fractions were effective against all cell lines, but showed the highest cytotoxicity against P338, HL60 and LL2 (IC_50_ < 10–62.4 µg/mL). Sub-fractions showed cytotoxicity against all cell lines, with IC_50_ values between < 10–91.2 μg/mL.	[44]
*P. chinensis*	Corilagin and ellagic acid	WP	SiHa	Effective dose-dependent activity, with cell inhibition of 59 and 81% (at a concentration of 100 µM) and IC_50_ values of 21.5 and 28.7 µM, respectively.	[54]
*P. decipiens*	Phenolic acids and flavonoids (isolated from MeOH extract)	L, S	CaCo-2, PC3	IC_50_ = 0.5–1.1 µg/mL.	[58]
*P. glabra*	(-)-pinocembrin (isolated from MeOH extract)	AP	THP-1, A549, PANC-1, HeLa, MCF7	IC_50_ values between 1.88 to 11.00 mg/mL.	[63]
*P. hydropiper*	MeOH extract and its fractions (saponins, Chl and EtOAc)	WP	Potato disc anti-tumour assay	Assays were performed on Agrobacterium tumefaciens containing tumour-inducing plasmid. Tumour inhibitions between 80 to 90% (at a dose of 1000 µg/mL) and IC_50_ values between 18.39 and 342.53 µg/mL.	[83]
	β-sitosterol and stigmasterol (isolated from Chl and EtOAc fractions)	AP	NIH/3T3, HeLa, MCF-7	Effective dose-dependent activity, with percentage of cytotoxicity between 67.05 and 87.5% and IC_50_ of 170–425 µg/mL (for β-sitosterol) and 60–170 µg/mL (for stigmasterol).	[84]
	MeOH extract	AP	EAC	Cell growth inhibition of 84.54% (at a concentration of 50 mg/kg/day) and improved at a 68% the survival of mice.	[85]
		L		Significantly decreased tumour volume, packed cell volume and viable tumour cell and increased non-viable tumour cell. At a dose of 100 mg/kg, the median survival time (MST) was 37.21, respect to reference bleomycin (46.60).	[86]
*P. lapathifolia*	Pinostrobin (isolated from petroleum ether extract)	AP	Jurkat and HL60	Dose-dependent effects, with a percentage of apoptotic cells > 70% (at a dose of 1 µM) and a percentage of necrotic cells > 80% (at a dose of 10 µM).	[99]
	Vanicoside B and lapathoside A (extracted with MeOH)		EBV-EA	Carcinogenesis was induced by 7,12-dimethylbenz[a]anthracene (DMBA, as initiator), (*E*)-methyl-2-[(*E*)-hydroxyamino]-5-nitro-6-methoxy-3-hexenamide (NOR-1) and 12-*O*-tetradecanoylphorbol-13-acetate (TPA) as a promoter. The number of papillomas per mouse were reduced at 3.4 and 2.6 after 15 weeks compared to the control (DMBA + TPA, 9.1 papillomas per mouse). In the NOR-1 + TPA treatment, vanicoside B reduced the number of papillomas from 7.2 (control) to 3.	[100]
*P. limbata*	Cardamomin and 2′,4′-dihydroxy-3′,6′-dimethoxychalcone (isolated from MeOH extract)	AP	MCF-7, THP-1, PC-3, HeLa	Proliferation inhibition > 50%. The best activity was observed against THP-1 cell line, with IC_50_ < 4 µg/mL.	[102]
	Flavonoids	L	CCRF-CEM, MDA-MB-231-pcDNA3, HCT116, U87MG, HepG2	The flavonoid 4-hydroxy-2,6-dimethoxychalcone, showed the best activity against all cell lines, with IC_50_ values of 9.37, 19.58, 6.8, 35.25 and 58.63 µM. The flavonoids cardamomin and 2,4-dihydroxy-3,6-dimethoxychalcone were effective against CCRF-CEM cell line (IC_50_ of 8.59 and 10.67 µM, respectively).	[103]
*P. minor*	EtOAc (100%), aqueous-EtOH (50%), MeOH (100%), EtOH (70%) extracts and aqueous solution (100%)	L	HT-29, HCT-116, CT-26	EtOAc extract (100%) showed the highest cytotoxic effect against HCT-116 and CT-26 and Aq-EtOH extract (50%) was the most effective against HT-29 cell line (IC_50_ of 7.0 and 24.0 µg/mL, respectively). Aq solution (100%), MeOH (100%) and EtOH (70%) extracts showed moderate activity, with IC_50_ of 34.0–78.0 µg/mL (HT-29), 13.0–33.0 µg/mL (HCT-116) and 20.0–29.0 µg/mL (CT-25), respectively. IC_50_ values for standard agent Doxorubicin: 0.63 (HT-29), 0.46 (HCT-116) and 0.14 µg/mL (CT-26).	[116]
	Polygonumins A (extracted with MeOH)	St	K562, MCF7, HCT116	Good activity compared to the positive control Doxorubicine (IC_50_ = 2.25–3.24 and 0.52–2.97 µg/mL, respectively).	[117]
*P. orientalis*	EtOAc and *n*-But extracts	F	HeLa, SMMC-7721	Proliferation inhibition about 40–60% at doses between 50 to 450 µg/mL and after 48 h.	[128]
	EtOAc and Acetone extracts	Fr	95D	IC_50_ values = 199.1 mg/L (for EtOAc extract) and 261.2 mg/L (for acetone extract).	[129]
*P. senegalensis*	Phenolic acids and flavonoids (isolated from MeOH)	L, S	CaCo-2, PC3	IC_50_ 1.5–3.5 µg/mL.	[58]
*P. stagnina*	EtOAc and *n*-Hex extracts	AP	Potato assay disc	Moderate activity, with inhibition of tumour growth between 50 to 78.6% at a concentration of 200–400 µg/disc and IC_50_ values = 180 and 200 µg/disc (for EtOAc and *n*-Hex extracts, respectively). Positive control: vincristine sulphate 3.125 µg/disc, 100% inhibition tumour).	[37]
*P. tinctoria*	MeOH and EtOH extracts	F, L, St, S	HEK 293, HCT-116, HeLa, Hep3B, MCF-7, SNU-1066, SNU-601	EtOH extract from flowers was the most effective against all cell lines, except for MCF-7 and SNU-601, with survival rate of cancer cells = 5.10–25.27%. Leaves’ MeOH extract showed a good activity against HCT-116, HeLa, Hep3B and SNU-1066 (survival cancer cells = 6.89–26.47%); stems’ MeOH extract was effective against Hep3B and SNU-1066 (survival cancer cells = 26.94–29.28%); and seeds’ MeOH extract showed a good activity against HEK 293, HeLa, Hep3B and SNU-601 (survival cancer cells = 22.89–29.85%).	[150]
	Tryptanthrin (isolated from EtOAc extract)	L	U-937, HL-60	Compound showed 100% of cytocidal effects on both cell lines (at a concentration of 6.3 mg/mL) and inhibited DNA synthesis at dose-dependency.	[152]
*P. aviculare*	MeOH extract	AP	MCF-7	The extract induced cytotoxicity in MCF-7 cell line, with a 99% of cell death at the concentration of 400 ng/µl after 24 hrs.	[161]
			Hela-S	Showed cytotoxic effect at IC_50_ values between 0.27 and 0.41 mg/mL and caused complete apoptosis at 24 h of treatment.	[162]
*P. bellardii*	MeOH, EtOAc and *n*-But extracts and isolated compounds (gallic acid, quercetin/its derivatives and myricetin/its derivatives)	AP	HeLa, MCF-7, HepG-2	At a concentration between 130 and 170 µg/mL, all extracts and fractions inhibited cell viability at 20% of all cell lines tested. *n*-But extract was the most potent against three cell lines (IC_50_ = 15.26, 50.66 and 30.09 µg/mL, respectively); MeOH and EtOAc extracts showed a good activity against HeLa cells (IC_50_ = 48.6 and 44.14 µg/mL, respectively); and quercetin derivatives and myricetin and its derivatives were effective against HepG-2 (IC_50_ between 41.03 to 70.77 µg/mL).	[168]
*P. cognatum*	EtOH extract	L	MDA-MB-231	MDA-MB-231 cell viability < 50% and IC_50_ = 0.053 mg/mL.	[169]
*P. perfoliatum*	8-oxo-pinoresinol (isolated from MeOH extract)	T	Bcap-37, SMMC-7721, K562, RKO, PC3	IC_50_ values from 8.32 to 30.1 µg/mL (positive control Mitomycin, IC_50_ values from 1.75 to 6.24 µg/mL).	[20]
	EtOAc extract	AP	Smur180, SGC-7901, Colo320, PC-3, HL60	Smur180 inhibition cells = 58.46% (at a dose of 200 mg/kg) and inhibited the growth and proliferation of other cell lines, with IC_50_ < 50 µg/mL.	[185]
		AP	PANC-1, PC-3, SGC-7901, BT-325, HepG2, A549, Hela	Inhibition cell lines = 70.1–90% and IC_50_ values between 20.6 to 40.7 µg/mL. Furthermore, the extract arrested cells at G2 phase, increased the proliferation of T and B lymphocytes, promoted the activities of NK and cytotoxic T lymphocytes (CTLs) and induced cell apoptosis.	[186]
*P. thunbergii*	Isorhamnetin	AP	NIH3T3, SW620	The compound decreased the percentage of cell proliferation with IC_50_ values of 4.1 µg/mL (for NIH3T3) and 22.4 µg/mL (for SW620).	[192]
*P. viscosum*	MeOH crude extract	B	Brine shrimp lethality test	The extract showed a good cytotoxic ability respect to the standard Vincristine sulphate (IC_50_ = 6.34 and 0.825 µg/mL, respectively).	[194]
	Quercetin 3-*O*-(6-feruloyl)-β-d-galactopyranoside	WP	OVCAR-3	IC_50_ = 13.33 µg/mL. Authors considered that this effect could be attributed to the presence of moieties such as quercetin, galactosyl and principally feruloyl in this compound.	[195]

**Table 6 molecules-26-05956-t006:** Antiviral activities of *Persicaria* and *Polygonum* species. Extracts, compounds, part of the plant used, target virus and references are shown. Extracts: EtOH (ethanol); MeOH (methanol); Hex (hexane); EtOAc (ethyl acetate); But (butanol). Part used: L (leaves); F (flowers); Fr (fruits); R (roots); Sp (sprouts); S (seeds); St (stems); Rz (rhizomes); T (tubers); B (barks); WP (whole plant); AP (aerial parts).

Species	Extract/Isolated Compounds	Part Use	Pathogen	Summarized Bioactivity	Ref.
*P. chinensis*	MeOH, But and EtOAc extracts.	WP	H3N2-HK, H1N1-PR8, Lee	EC_50_ = 18.3–38.4 µg/mL (for H3N2) and 45.9–70.1 µg/mL (for H1N1and Lee).	[55]
	Ellagic acid, methyl gallate and caffeic acid (isolated from EtOAc extract)			Significantly inhibited viral replication (EC_50_ = 14.7–81.1 µg/mL) by suppressing virus replication in cells.	
*P. ferruginea*	EtOH extract	AP	HHV-1, DEN-2, VACV-WR	EC_50_ values = 21.1, 24.6 and 34.2 µg/mL, respectively.	[60]
*P. glabra*	2-methoxy-5-oxo-2,5-dihydrofuran-3-yl (2*E*)-(−)-3-phenylprop-2-enoate (isolated from MeOH extract)	AP	HIV-1VB59, HIV-1UG070	IC_50_ values = 15.68–22.43 mg/mL.	[63]
*P. lapathifolia*	Lapathoside A and D, vanicoside B and hydropiperoside (isolated from MeOH extract)	AP	EBV-EA	Inhibition of activation on EBV-EA higher than 85, 60 and 30%.	[100]
*P. minor*	Polygonumins A	St	HIV-1	Inhibition of HIV-1 protease = 56.51% (positive control Pepstatin A, inhibition = 81.48%). The authors considered that the phenyl propanoid glycoside moiety present in Polygonumins A, may be the responsible for the anti-HIV protease activity.	[117]
	EtOH extract	L	HSV-1, VSV	MIC values = 0.01 and 0.02 mg/mL (for HSV-1 and VSV, respectively).	[118]
*P. punctata*	Aqueous extract	AP	HSV-1, RSV	ED_50_ values = 169.7 and 120 µg/mL (for HSV-1 and RSV, respectively)	[141]
	MeOH	WP	HSV-1	At a MIC of 20 µg/mL, the extract caused the complete virus inactivation.	[138]
*P. tinctoria*	MeOH and EtOAc extracts	WP	H1N1-PR8, HSV-1, VV, MCMV, VSV	IC_50_ values between 1.25 and 16.6 µg/mL (for MeOH extract) and 0.63 to 50 µg/mL (for EtOAc extract).	[153]
	Aqueous extract	L	HIV-1	Inhibition of HIV-1 (IIIB) infection at EC_50_ value of 0.5 µg/mL.	[154]
*P. perfoliatum*	Quercetin-3-*O*-β-d-glucuronide	WP	influenza A virus	Inhibition of 27.94%, at a concentration of 6 mg/kg (reference drug Ribavirin, inhibition of 23.97%).	[184]
	Flavonoids (extracted with MeOH and *n*-Hex)	L	HSV-1	At a concentration of 62.5 µg/mL, flavonoids reduced more than 80% the number of plaques in infected cultures, in the same way as the control ACV. At a dose of 30 mg/kg/day, the survival and mean survival time (MST) of mice induced with encephalitis HSV-1 were 80% and 19.0 days, respectively, higher values compared with the control ACV (survival of 70% and MST = 18.5 days).	[187]
	Gallic acyl groups (isolated from *n*-But and water extracts)	WP	HBV	Inhibition up to 74% secretion of antigen (HBeAg).	[188]
*P. viscosum*	Quercetin 3-*O*-(6-feruloyl)-β-d-galactopyranoside and viscoazulone	WP	HIV-1	Reverse transcriptase inhibitory activity with IC_50_ values of 33.13 and 25.61 mg/mL (for quercetin and viscoazulone, respectively).	[195]

**Table 7 molecules-26-05956-t007:** Antiparasitic activities of *Persicaria* and *Polygonum* species. Extracts, compounds, part of the plant used, antiparasitic activity and references are shown. Extracts: EtOH (ethanol); MeOH (methanol); Hex (hexane); EtOAc (ethyl acetate). Part used: L (leaves); F (flowers); Fr (fruits); R (roots); Sp (sprouts); S (seeds); St (stems); Rz (rhizomes); T (tubers); B (barks); WP (whole plant); AP (aerial parts).

Species	Extract/Isolated Compounds	Part Use	Pathogen	Summarized Bioactivity	Ref.
Anti-malarial					
*P. acuminata*	MeOH extract	L	*P. falciparum*	IC_50_ = 8 µg/mL	[26]
*P. glabra*	EtOH extract	L	*Plasmodium* sp.	IC_50_ = 6.6 µg/mL	[66]
Anti-trypanocide					
*P. decipiens*	2,4-dimethoxy-6-hydroxychalcone (chalcone 1) and 2,5-dimethoxy-4,6-dihydroxychalcone (chalcone 2)	AP	*T. brucei* *T. congolense*	EC_50_ = 1.8–8.8 µM (for chalcone 1) and 13.9–34 µM (for chalcone 2). Positive controls: Diminazene (EC_50_ = 0.15–1.43 µM) and Pentamidine (EC_50_ = 0.0034–0.72 µM).	[59]
*P. ferruginea*	*n*-Hex, DCM and EtOAc extracts	AP	*T. cruzi* *T. brucei*	IC_50_ values = 8.6–10.5 µg/mL (for Hex and dichloromethane extracts) and 50–90 µg/mL (for EtOAc extract). Positive controls: Pentamidine (IC_50_ = 6.4 and 2.2 µg/mL) and Benznidazole (IC_50_ = 34.7 and 54.1 µg/mL).	[61]
	2′-hydroxy-4′,6′-dimethoxychalcone, flavokawin B (1), 2′,6′-dihydroxy-3′,4′-dimethoxychalcone, pashanone (2) and (2′,4′- dihydroxy-6′-methoxychalcone, cardamonin or alpinetin chalcone (3) (isolated from Hex extract)			Compounds 1, 2 and 3 were effective against *T. cruzi* (IC_50_ between 9.5 and 32.3 µM). Against *T. brucei*, only compound 1 was active, with IC_50_ = 6.2 (for strain 427) and 4.8 µM (for strain 29–13).	
	MeOH extract		*T. cruzi*	IC_50_ = 37 µg/mL.	[26]
*P. hydropiper*	Cardamomin, vanicoside F, ketopinoresinol, isorhamnetin and pinosylvin (isolated from DCM soluble portion)	WP	*T. brucei*	IC_50_ = 0.49–0.8 µg/mL (for cardamomin) and 0.49–7.77 µg/mL (for other compounds). Positive control: α-difluoromethylornithine (DFMO), IC_50_ = 3.02 µg/mL.	[87]
Anti-leishmanial					
*P. glabra*	Aqueous extract	WP	*Leishmania tropica*	At a minimal concentration tested (0.05 µg/mL), extract showed a 4.23% parasite mortality, while using a concentration of 50 µg/mL, the mortality raised up to 68.1%.	[67]
*P. hydropiperoides*	MeOH extract	F	*Leishmania amazonensis*	IC_50_ = 73 µg/mL.	[95]
Anthelmintic					
*P. alpina*	MeOH extract	Rz	Earthworms	Earthworms’ death times = 71.4 min at a concentration of 100 mg/mL (standard Albendazole, death time = 56.6 min).	[27]
*P. chinensis*	Aqueous and MeOH extracts	L	*Pheretima posthuma*	Inhibition and death at 5.83 and 16.5 min (for Aqueous extract), respectively, and at 9.25 and 19.67 min (for MeOH extract), respectively, at a dose = 100 mg/mL. Standard drug: Albendazole (inhibition and death at 5.33 and 6.92 min).	[56]
*P. hydropiper*	MeOH extract	AP	*P. posthuma*	At the extract concentration of 50 mg/mL, the time of paralysis and death of earthworms were 12.44 and 18.19 min. Positive control: Piperazine citrate, 10 mg/mL (time of paralysis and death = 24 and 38 min, respectively).	[85]
	Saponins, Chl and *n*-But (fractionated from MeOH extract)	WP	*P. posthuma*	Paralysis time between 8 and 11 min and death time between 50 and 66.33 min at a concentration of 10 mg/mL.	[88]
*P. lapathifolia*	MeOH extract	St	*P. posthuma*	At a concentration of 60 mg/mL, the times taken for paralysis and death were 9 and 23.66 min, respectively. Standard drug: Piperazine citrate, 10 mg/mL (time taken for paralysis and death = 9.33 and 36 min, respectively).	[101]
*P. muricatum*	EtOH extract	L	Earthworms	At a concentration of 100 mg/mL, the paralysis and death times were 35 and 43 min, respectively. Standard drug: Albendazole (paralysis and death time = 20 and 30 min, respectively). Phytochemical analysis identified various constituents such as alkaloids, carbohydrates, glycosides, phytosterols, phenolic compounds, tannins, saponins, proteins and amino acids.	[179]
*P. viscosum*	MeOH extract				[194]

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
