# Peer review of "An Update on Phytochemicals and Pharmacological Activities of the Genus *Persicaria* and *Polygonum"

_molecules, 2021, doi:10.3390/molecules26195956_

Round 1

Reviewer 1 Report

The paper is a valuable overview. My concern is the chemical nomenclature. So, (-) Pinocembrin should be written with minus and hyphen, as (−)-Pinocembrin.

Name (-) (2)-2-methoxy-2-butenolide-3-cinnamate is not any good. The structure named also butenolide cinnamate shall be named as 2-methoxy-5-oxo-2,5-dihydrofuran-3-yl (2E)-(−)-3-phenylprop-2-enoate.

Quercetin-3-O-glucoside shall be names as quercetin 3-O-β-D-glucopyranoside, where -D- is lowercase capital letter.

Where atom symbol serves as locant, it shall be in italics as -O-.

Names as quercetin-3-O-galactoside, quercetin-3-O-rhamnoside are not showing the configuration on sugar C-1.

Stereodescriptors D and L e.g. in  L-ascorbic acid shall be in lowercase capital letter. Steteodescriptos E, Z, R and S shall be in italics, uppercase.

Expression "unidentified (5, 6-dihydropyranobenzopyrone)" is not correct, authors in original paper are saying "hitherto unidentified compd., 5,​6-​dihydropyranobenzopyrone". Moreover, I was not able to get the original paper in Zhongguo Tianran Yaowu so I was not able to correct the name "5, 6-dihydropyranobenzopyrone", which to me does not well represent a compound.

In 3-O-(6-feruloyl)-b-D-galactopyranoside -b- should be beta.

My remarks to the nomenclature shall be understood even if the original cited paper uses the criticized name.

Otherwise, as I wrote the paper is valuable and well written and after the nomenclature is corrected, it is worth publishing.

Author Response

Responses to Reviewer 1:

The paper is a valuable overview. My concern is the chemical nomenclature. So,

(-) Pinocembrin should be written with minus and hyphen, as (−)-Pinocembrin.

Response: corrected.

Name (-) (2)-2-methoxy-2-butenolide-3-cinnamate is not any good. The structure named also butenolide cinnamate shall be named as 2-methoxy-5-oxo-2,5-dihydrofuran-3-yl (2E)-(−)-3-phenylprop-2-enoate.

Response: corrected.

Quercetin-3-O-glucoside shall be named as quercetin 3-O-β-D-glucopyranoside, where -D- is lowercase capital letter.

Response: corrected.

Where atom symbol serves as locant, it shall be in italics as -O-.

Response: corrected throughout the text.

Names as quercetin-3-O-galactoside, quercetin-3-O-rhamnoside are not showing the configuration on sugar C-1.

Response: Names were corrected.

Stereodescriptors D and L e.g. in L-ascorbic acid shall be in lowercase capital letter. Steteodescriptos EZR and S shall be in italics, uppercase.

Response: these errors were corrected.

Expression "unidentified (5, 6-dihydropyranobenzopyrone)" is not correct, authors in original paper are saying "hitherto unidentified compd., 5,​6-​dihydropyranobenzopyrone". Moreover, I was not able to get the original paper in Zhongguo Tianran Yaowu so I was not able to correct the name "5, 6-dihydropyranobenzopyrone", which to me does not well represent a compound.

Response: we modified the sentence taking into account that mentioned in the original paper: “On the other hand, Tantry et al. [76] isolated ten compounds from EtOH extract of P. amplexicaulis, nine known compounds and a previously undetermined one (5, 6-dihydropyranobenzopyrone) and tested its antioxidant activity”.

In 3-O-(6-feruloyl)-b-D-galactopyranoside -b- should be beta.

Response: corrected.

My remarks to the nomenclature shall be understood even if the original cited paper uses the criticized name.

Otherwise, as I wrote the paper is valuable and well written and after the nomenclature is corrected, it is worth publishing.

Response: we wish to acknowledge reviewer 1 for the suggestions and kindly comments.

Reviewer 2 Report

The article presents the overview of biological activities of genera Persicaria and Polygonum. The work is based on articles from last 20 years, however the available reviews on the topic are published in 2015, 2018… hence the novelty of the work is not justified to the extent.

General remarks

Tables are prepared very well. They are clear and comprehensive.

The limitation of the study is lack of any comment on the subject, therefore it has to be said that work presents description instead of discussion of many bioactivities of selected plant species.

It would be good to correlate the activity with secondary metabolites, it would be good to mention limitations of performed studies, to add what can be done next to validate the activities in animals and humans…

Also to conclude which species are the most promising and at the same time widely available in many countries.

Other remarks

Lines 77-81: aldehydes is chemical group, which is usually present in every group of plant sec metabolites, terpenoids are subclass of terpenes, flavonoids and tanins belong to polyphenolic compounds – seems that authors got lost in the chemistry of natural products, this has to be clarified

Table 1. no sources provided – no referenes given

Authors had given no comments on antimicrobial activity  - the table is extensive however the review should also present some opinion and perspective, this is missing.

The next paragraph is also extensive, however this time the reader find the long text instead of table, this makes finding the information very difficult, definitely it would be much better to present these information as a table. But the authors should not forget to add their comment, which now is missing.

Similar situation is for Analgesic and anti-inflammatory activity; the reader find more than two pages of plain taxt to go through, and this is not a commentary but the description of performed studies.

Author Response

Responses to Reviewer 2:

General remarks

Tables are prepared very well. They are clear and comprehensive.

The limitation of the study is lack of any comment on the subject, therefore it has to be said that work presents description instead of discussion of many bioactivities of selected plant species.

Response: the aim of this review was to summarized and compiled the widespread information available for these two plant genera. Moreover, some discussions of the main bioactivities were added in the revised version (lines 101-116; 171-184; 339-354; 564-572; 581-584 and 705-710).

It would be good to correlate the activity with secondary metabolites, it would be good to mention limitations of performed studies, to add what can be done next to validate the activities in animals and humans…

Response: these suggestions were added in the discussion mentioned above (lines 101-116; 171-184; 339-354; 564-572; 581-584 and 705-710).

Also, to conclude which species are the most promising and at the same time widely available in many countries.

Response: The most promising species taking into account the highest number of medicinal properties found in the literature were expressed in lines 1059-1066 (conclusion). Unfortunately, the availability of the species in many countries is impossible for us to accede.

Other remarks

Lines 77-81: aldehydes are chemical group, which is usually present in every group of plant sec metabolites, terpenoids are subclass of terpenes, flavonoids and tannins belong to polyphenolic compounds – seems that authors got lost in the chemistry of natural products, this has to be clarified.

Response: These errors of classification were corrected.

Table 1. no sources provided – no references given.

Response: Table 1 is introductory to present the plant species. References are suitably provided in the following tables according to each bioactivity developed.

Authors had given no comments on antimicrobial activity - the table is extensive however the review should also present some opinion and perspective, this is missing.

Response: comment on antimicrobial activities of table 2 were included in lines 101-116.

The next paragraph is also extensive, however this time the reader finds the long text instead of table, this makes finding the information very difficult, definitely it would be much better to present this information as a table. But the authors should not forget to add their comment, which now is missing.

Response: all the text corresponding to antioxidant activities was deleted and a table was added instead, also final comments.

Similar situation is for Analgesic and anti-inflammatory activity; the reader finds more than two pages of plain text to go through, and this is not a commentary but the description of performed studies.

Response: a table and final comments were added instead of the text that refers to analgesic and anti-inflammatory activities.

Round 2

Reviewer 2 Report

The manuscript was substantially improved, however Table 1 is still missing references. To make it easier, authors can reffer to group of references used to prepare this table, but it has to be supported by literature sources.

Author Response

Response to reviewer 2 (second round)

The manuscript was substantially improved; however, Table 1 is still missing references. To make it easier, authors can refer to group of references used to prepare this table, but it has to be supported by literature sources.

Response: we acknowledge to reviewer 2 and we inform that Table 1 was referenced using the group of references used to prepare it.